# Dopamine D2Rs coordinate cue-evoked changes in striatal acetylcholine levels

Kelly M Martyniuk[1,2], Arturo Torres-Herraez[2,3], Daniel C Lowes[2,3], Marcelo Rubinstein[4], Marie A Labouesse[2,5,6], Christoph Kellendonk[2,3,7]*

[1]Department of Neuroscience, Columbia University, New York, United States; [2]Department of Psychiatry, Columbia University, New York, United States; [3]Division of Molecular Therapeutics, New York State Psychiatric Institute, New York, United States; [4]Instituto de Investigaciones en Ingeniería Genética y Biología Molecular, Consejo Nacional de Investigaciones Científicas y Técnicas, and Departamento de Fisiología, Biología Molecular y Celular, Facultad de Ciencias Exactas y Naturales, Universidad de Buenos Aires, Buenos Aires, Argentina; [5]Department of Health Science and Technology, ETH Zurich, Schwerzenbach, Switzerland; [6]Zurich Neuroscience Center, Winterthurerstrasse, Zurich, Switzerland; [7]Department of Molecular Pharmacology and Therapeutics, Columbia University, New York, United States

*For correspondence:
ck491@cumc.columbia.edu

Competing interest: The authors declare that no competing interests exist.

**Abstract** In the striatum, acetylcholine (ACh) neuron activity is modulated co-incident with dopamine (DA) release in response to unpredicted rewards and reward-predicting cues and both neuromodulators are thought to regulate each other. While this co-regulation has been studied using stimulation studies, the existence of this mutual regulation in vivo during natural behavior is still largely unexplored. One long-standing controversy has been whether striatal DA is responsible for the induction of the cholinergic pause or whether DA D2 receptors (D2Rs) modulate a pause that is induced by other mechanisms. Here, we used genetically encoded sensors in combination with pharmacological and genetic inactivation of D2Rs from cholinergic interneurons (CINs) to simultaneously measure ACh and DA levels after CIN D2R inactivation in mice. We found that CIN D2Rs are not necessary for the initiation of cue-induced decrease in ACh levels. Rather, they prolong the duration of the decrease and inhibit ACh rebound levels. Notably, the change in cue-evoked ACh levels is not associated with altered cue-evoked DA release. Moreover, D2R inactivation strongly decreased the temporal correlation between DA and ACh signals not only at cue presentation but also during the intertrial interval pointing to a general mechanism by which D2Rs coordinate both signals. At the behavioral level D2R antagonism increased the latency to lever press, which was not observed in CIN-selective D2R knock out mice. Press latency correlated with the cue-evoked decrease in ACh levels and artificial inhibition of CINs revealed that longer inhibition shortens the latency to press compared to shorter inhibition. This supports a role of the ACh signal and it's regulation by D2Rs in the motivation to initiate actions.

## Editor's evaluation

The study addressed interactions between two key striatal transmitters dopamine and acetylcholine during an appetitive behavioral task. Helping to reconcile conflicting evidence in the literature, the data show that changes in both transmitters are correlated and that decreases in acetylcholine with reward and reward cues are only partially a consequence of elevated dopamine release acting at D2 dopamine receptors on striatal cholinergic interneurons. This manuscript will be of interest to those interested in the neural correlates of appetitive behavior and dopamine and striatal function.

## Introduction

Dopamine (DA) plays a key role in learning, serving as a teaching signal that reflects reward prediction error (*Day et al., 2007*; *Mohebi et al., 2019*; *Nasser et al., 2017*; *Schultz et al., 1997*; *Steinberg et al., 2013*). This teaching function is encoded in the phasic bursting of DA neurons, which induces a rapid but transient increase of extracellular DA. DA is initially released in response to an unpredicted reward, but with learning the response shifts away from the reward outcome toward reward-predicting cues (*Schultz, 2007*; *Schultz et al., 1997*).

Like DA neurons, cholinergic interneurons (CINs) in rodents and their presumed counterparts, 'tonically active neurons' (TANs), in primates modulate their activity in response to reward-predicting cues and salient outcomes. CINs represent about 1–2% of the neurons in the striatum and regulate mental processes including reinforcement learning, action selection, associative learning, and cognitive flexibility (*Aoki et al., 2015*; *Bradfield et al., 2013*; *Joshua et al., 2008*; *Matamales et al., 2016*; *Maurice et al., 2015*; *Morris et al., 2004*; *Okada et al., 2014*). Pharmacogenetic inhibition of CINs in the nucleus accumbens (NAc) also increases the influence of appetitive cues on instrumental actions pointing to a role of striatal acetylcholine (ACh) in motivation (*Collins et al., 2019*). CINs are tonically active and show a multiphasic response to salient and conditioned stimuli that can include a short excitation followed by a prominent pause and rebound excitation (*Aosaki et al., 1994a*; *Aosaki et al., 1994b*; *Apicella, 2007*; *Apicella et al., 2009*; *Apicella et al., 2011*). This multiphasic response in CIN firing coincides with phasic activation of midbrain DA neurons that terminate in the striatum (*Joshua et al., 2008*; *Morris et al., 2004*; *Schultz, 2007*; *Schultz et al., 1997*). Furthermore, there is increasing evidence that DA and ACh regulate each other within the striatum (*Cachope and Cheer, 2014*; *Cachope et al., 2012*; *Chuhma et al., 2014*; *Cragg, 2006*; *Helseth et al., 2021*; *Kharkwal et al., 2016*; *Straub et al., 2014*; *Sulzer et al., 2016*; *Threlfell et al., 2012*; *Yan and Surmeier, 1991*).

Here, we will focus on the DA regulation of the multiphasic ACh response. One long-standing discussion in this regard has been whether the cholinergic pause is dependent on DA via DA D2R receptor (D2R) mediated inhibition of CINs. Early evidence that the CIN pause is DA-dependent originate from studies in non-human primates (NHPs). In vivo electrophysiological recordings from TANs have revealed a pronounced pause in firing to a reward-predicting stimulus. This pause was entirely abolished by 1-methyl-4-phenyl-1,2,3,6-tetrahydropyridine lesions of DA neurons and local administration of a D2R antagonist (*Aosaki et al., 1994b*; *Watanabe and Kimura, 1998*). Consistent with this, more recent slice physiology studies in rodents have shown that pauses in CIN activity can be induced by local application of DA or DA terminal stimulation, in which both are eliminated by pharmacological blockade of D2Rs (*Augustin et al., 2018*; *Chuhma et al., 2014*; *Straub et al., 2014*; *Wieland et al., 2014*). Additionally, optogenetic stimulation of NAc DA terminals results in a pause in CIN firing and this pause is prolonged when D2Rs are selectively overexpressed in CINs (*Gallo et al., 2022*). Lastly, pauses generated by DA or local stimulation of the striatum are eliminated in a selective CIN D2 knockout mouse (*Augustin et al., 2018*; *Kharkwal et al., 2016*). Taken together, the slice physiology experiments provide evidence that the CIN pause can be induced by DA activation in a CIN D2R-dependent manner while the NHP studies show the necessity for DA and D2Rs for the generation of the pause.

However, more recent evidence suggests that the CIN pause is not induced by DA but by cortical, thalamic, or long-range GABAergic inputs (*Brown et al., 2012*; *Cover et al., 2019*; *Ding et al., 2010*; *Doig et al., 2014*; *English et al., 2012*; *Matsumoto et al., 2000*; *Zhang et al., 2018*). Consistent with this, stimulation of cortical and thalamic inputs to the striatum in slices or in vivo induces a triphasic cholinergic pause. One model suggests that the cholinergic pause is generated by intrinsic properties of CINs. When CINs come out of the early glutamatergic excitation, voltage-gated potassium channels (Kv7.2/7.3) open and induce an after-hyperpolarization that induces the pause. In this model DA plays a role in augmenting the intrinsically induced pause (*Zhang et al., 2018*). Consistent with this, thalamo-striatal stimulation induced a pause that was shortened but not fully abolished by a D2R antagonist (*Cover et al., 2019*). However, in earlier influential slice physiology experiments, the pause induced by thalamic stimulation was fully blocked by D2R antagonism suggesting that activation of DA release from intrastriatal DA terminals was responsible for pause generation (*Ding et al., 2010*).

One limitation of the mechanistic studies in rodents has been that they relied on stimulation experiments rather than on DA evoked by natural stimuli. While the early NHP studies suggested necessity for DA in inducing the pause during behavior, these studies lacked the cellular specificity for excluding

the possibility that the effects of pharmacological D2R blockage were due to inhibiting D2Rs on CINs vs other neuronal populations.

Here, we used genetically encoded biosensors (*Labouesse and Patriarchi, 2021*) to simultaneously monitor DA and ACh in the dorsal striatum during behavior in mice with pharmacological blockade and/or selective ablation of D2Rs from CINs. Using this approach, we addressed the question of whether the natural stimulus-induced pause is fully dependent on DA or not. We first determined whether changes in DA and ACh levels occur simultaneously to reward-predicting stimuli in mice as has been shown in NHPs via electrophysiological recordings of DA and TAN neurons (*Morris et al., 2004*). In vivo imaging of ACh and DA levels revealed cue-induced decreases in striatal ACh and increases in DA levels, confirming the ability to measure concomitant ACh dips and DA peaks with functional imaging. Using a Pavlovian learning task, we confirmed that both signals co-occur and develop in parallel during the training of the task. Using a simpler reinforcement task that enables better quantification of the neuromodulator signals, we quantified cue-induced changes in DA and ACh changes after manipulating D2R function. We found that selective ablation of D2Rs from CIN or blocking D2Rs in control mice with the selective D2R antagonist eticlopride did not abolish the stimulus-induced decrease in ACh levels. Rather it shortened the duration of the decrease and enhanced ACh rebound levels in a dose-dependent manner. This indicates that DA is necessary for controlling the overall shape of the ACh signal. During simultaneous recordings experiments, the relationship between DA and ACh was strongest in response to reward-predicting cues but still present during the intertrial interval (ITI) supporting a general mechanism by which DA coordinates ACh levels. At the behavioral

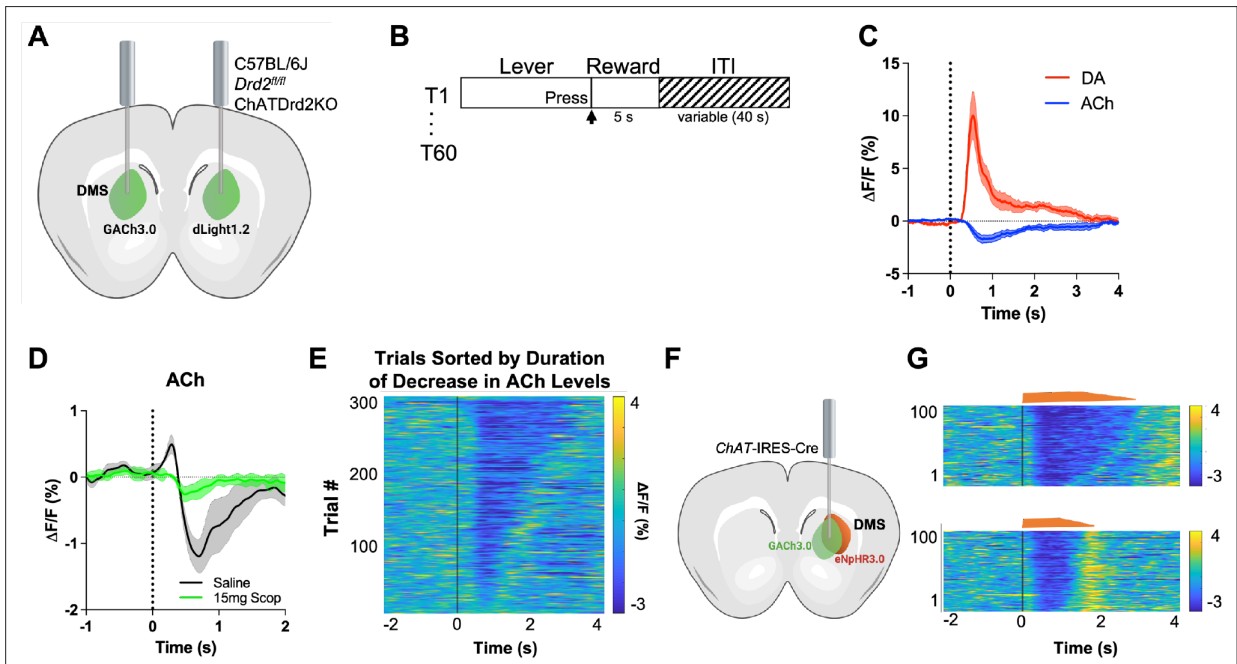

**Figure 1.** GACh3.0 reliably measures fast decreases in acetylcholine (ACh) during an instrumental task. (**A**) Schematic of the surgery setup. All mice were injected with both biosensor viruses (GACh3.0 and dLight1.2) in separate hemispheres of the dorsal medial striatum (DMS) and counterbalanced across mice. Fiber photometry lenses were bilaterally implanted at the site of viral injection to simultaneously monitor ACh and dopamine (DA) in the same mouse. (**B**) Continuous reinforcement (CRF) task design. Mice were trained to press a lever to retrieve a milk reward for 60 trials/day with a variable intertrial interval (ITI) (40 s). (**C**) Changes in fluorescence (ΔF/F[%]) aligned to lever extension (time point = 0 s). DA levels (red) increased and ACh levels (blue) decreased, N=5 mice in trained mice. (**D**) 15 mg/kg of scopolamine (green), an mAChR antagonist, blunts the initial ACh peak and dip compared to saline (black) confirming that the GACh3.0 sensor is reporting changes in ACh levels. N=4 mice. (**E**) Heatmap of ACh responses aligned to lever extension (time = 0 s) and sorted by the duration of ACh decrease for 300 individual trials (60 trials in 5 mice). (**F**) Schematic of the surgery setup. *ChAT*-ires-Cre mice were co-injected with GACh3.0 and Cre-dependent halorhodopsin into the DMS and a fiber photometry lens was implanted at the site of viral injection. (**G**) Approximation of trials with short dips (bottom) and long dips (top) using the short and long optogenetic inhibition protocol (100 trials, 20 trials/5 mice).

The online version of this article includes the following figure supplement(s) for figure 1:

**Figure supplement 1.** ACh levels in response to lever presentation at different recording location in the striatum.

level, D2R antagonism increased latency to lever press to a reward-paired lever, but this relationship was abolished when we inactivated CIN D2Rs. Moreover, cue-evoked changes in ACh levels correlated with the latency to press, and short artificial inhibition of CINs at lever extension regulated the latency to press. Altogether this supports a role of the cue-induced ACh signal in the motivation to initiate actions.

## Results

### GACh3.0 allows for measuring fast decreases in task-evoked ACh levels

First, we validated our experimental approach that uses fiber photometry and genetically encoded fluorescent indicators to simultaneously measure DA and ACh levels. Mice were imaged in separate hemispheres within the same animal as shown in *Figure 1A* during a continuous reinforcement (CRF) task (*Figure 1B*). We aligned our photometry signals to the lever extension, which with training becomes a reward-predicting cue. After 3 days of training, we observed an increase in DA (red) and a decrease in ACh (blue) at lever extension presentation (*Figure 1C*). To confirm that the fluorescent indicator, GACh3.0, is measuring changes in ACh levels (and not movement artifacts or electrical noise), we measured the GACh3.0 signal in the presence of 15 mg/kg scopolamine, a muscarinic antagonist, which targets the GACh3.0 parent receptor (M3R). We found that the competitive antagonist scopolamine abolished the early increase and the subsequent decrease in the fluorescent signal, indicating that GACh3.0 indeed quantifies ACh binding and thus surrounding ACh levels (*Figure 1D*).

To confirm that the GACh3.0 sensor has the kinetics to measure a rapid decrease in ACh levels, we expressed the inhibitory opsin eNpHR3.0 in *ChAT*-IRES-Cre mice to selectively inhibit CINs. Lever extension induced decreases in ACh levels within 250 ms (*Figure 1E*). Light activation of eNpHR3.0 in a home cage induced a decrease with even shorter latency (latency to decrease onset 206.4 [186.8–226.1] ms, n=5 mice), which was followed by a rebound in ACh levels (*Figure 1F*). The rebound is consistent with CINs displaying rebound activity after injecting hyperpolarizing currents in brain slices (*Wilson, 2005*) and optogenetic inhibition in vivo (*English et al., 2012*). These data show that GACh3.0 can measure fast decreases in ACh levels. It also indicates that ACh levels are tightly controlled by CIN neuron activity.

### Variability between animals

While analyzing the ACh signals we found that some mice showed an initial peak in ACh levels (*Figure 1D*) while others did not (*Figure 1C*). This has been described at the neuronal level when recording from individual neurons (*Apicella et al., 1997*; *Kimura et al., 1984*) but here it is observed at the level of ACh levels released by a population of neurons. While the origin of the between animal variability is unclear, we believe that it is related to the location of recording. Generally, more lateral/dorsal recording location showed an initial peak in ACh levels while more medial/ventral location did not show the initial peak (*Figure 1—figure supplement 1*). The origin for this variability should be addressed in a more systemic way in the future.

### Simultaneous development of DA and ACh signals in response to a reward-predicting stimulus

To determine whether changes in DA and ACh levels in response to reward-predicting stimuli are co-incident, we measured the release of DA and ACh during a Pavlovian reward learning task (*Figure 2A*). On day 1 of training, we observed an increase in DA (red) and a decrease in ACh (blue) during unexpected reward following the offset of the CS+ (*Figure 2C*). Over training, we saw these changes in both DA and ACh shift to the onset of the CS+ tone, while decreasing to the now expected reward. We did not observe these changes during CS− trials. We then related the changes in DA and ACh to changes in anticipatory head entries during the CS+ as a measure of learning. We found that both DA and ACh signals correlated well with anticipatory head entries in one animal (*Figure 2C*). However, other mice did not show any anticipatory responding as this task is non-contingent and head poking is not required to obtain the reward during CS+ trials. These findings indicate that DA and ACh signals co-develop with learning in response to a reward-predicting stimulus.

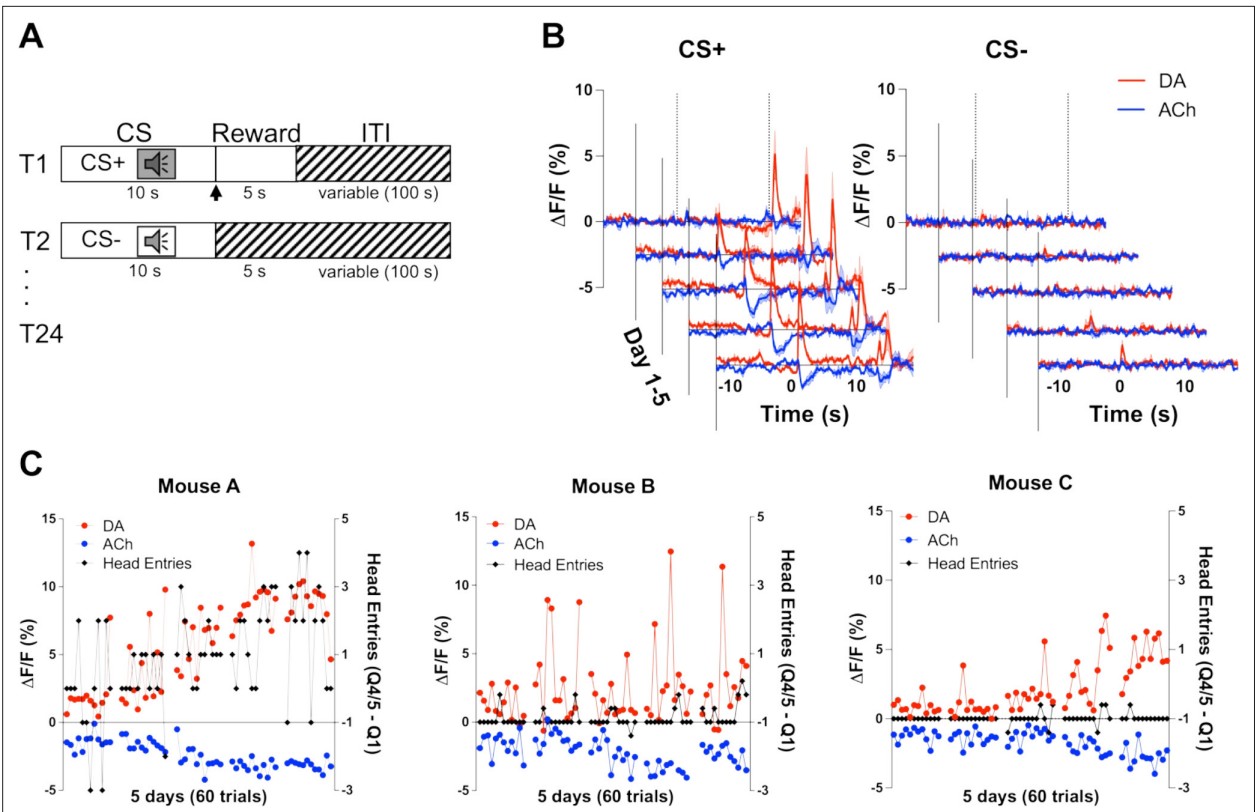

**Figure 2.** Co-development of dopamine (DA) and acetylcholine (ACh) signals to a reward-predicting cue. (**A**) Pavlovian task design. Mice were trained on 24 (12 CS+, 12 CS−) trials/day for 5 days. Each trial starts with a 10 s tone (CS+ or CS−). At the end of the CS+ a dipper comes up presenting a milk food reward for 5 s. There is an intertrial interval (ITI) variable in length (100 s). (**B**) Changes in fluorescence (ΔF/F [%]) over 5 days of training for DA (red) and ACh (blue) aligned to CS+ (left) and CS− (right) onset. Signals were averaged over 12 CS+ and 12 CS− trials/day, N=3 mice. (**C**) Maximum change in DA peak (blue) and ACh dip (red) after CS+ onset over 5 days of training (60 trials) for mouse A (left). Anticipatory responding (black) is calculated as the difference in nose poking during the CS+ quintile with the maximum responses (Q4 or 5) and the first quintile. Correlations between DA and ACh maxima and behavioral responding: $r=0.4$, $p<0.002$ and $r=−0.41$, $p<0.002$ in mouse A, respectively. Correlation between DA and ACh signals: $r=−0.7041$, $p<0.0001$. We did not observe the same correlation between DA/ACh and anticipatory responses in mouse B (middle) or mouse C (right). Correlation between DA and ACh signals: mouse B ($r=0.03997$, $p=0.7617$) and mouse C ($r=−0.6687$, $p<0.0001$).

## D2 receptor blockade dose dependently shortens the decrease and enhances the rebound in ACh levels

To determine if the cue-induced ACh decrease is dependent on DA activation of D2Rs, we used the CRF task as it allows for more trials per session aiding the quantification of the signal. After systemic delivery of the D2R antagonist eticlopride we found a dose-dependent shortening of the ACh decrease, which uncovered a rebound following the decrease (*Figure 3A*). We quantified these changes by calculating the area under the curve (AUC), dip duration and dip amplitude. We found that eticlopride significantly reduced the negative AUC (*Figure 3B*), increased the rebound AUC (*Figure 3C*), increased the total AUC (*Figure 3D*), and decreased the dip duration (*Figure 3E*), while the dip amplitude was not affected by D2R antagonism (*Figure 3F*). This suggests that D2Rs do not participate in the initial induction of the ACh decrease but do increase the duration of the decrease and prevent rebound activity following the ACh decrease. Since DA neurons are inhibited by D2 auto-receptors, we also analyzed the effect of D2R antagonism on cue-induced DA release and quantified changes in peak amplitude and AUC (*Figure 3—figure supplement 1A*). We found an overall effect of drug increasing the peak amplitude (*Figure 3—figure supplement 1B*) with the most prominent increase between saline and 0.1 mg/kg eticlopride. There was no overall effect of drug on the AUC (*Figure 3—figure supplement 1C*). These results confirm that blocking D2 auto-receptors on DA neurons increases phasic DA release.

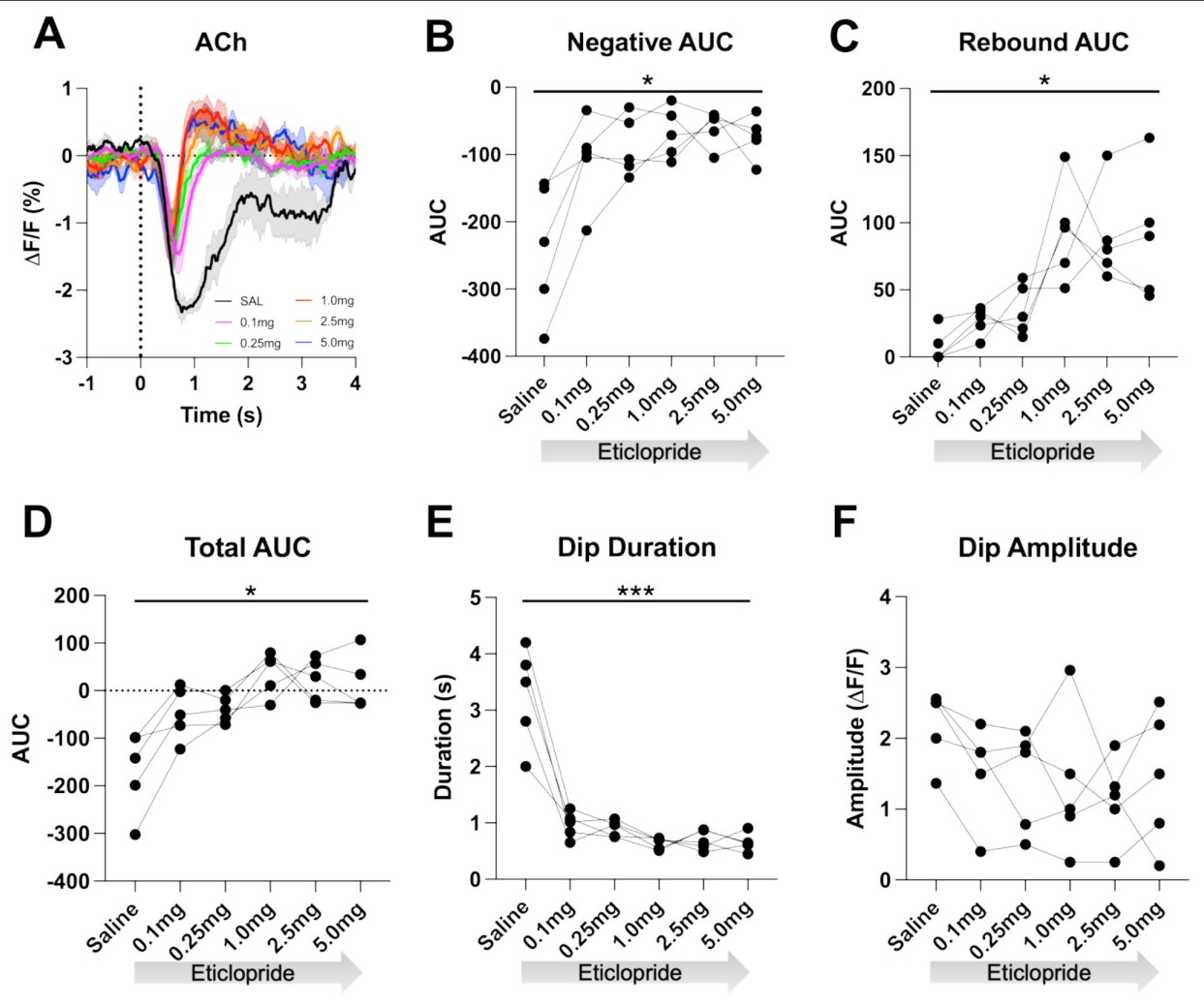

**Figure 3.** D2R antagonism decreases acetylcholine (ACh) dip duration and enhances rebound. (**A**) Changes in ACh fluorescence (ΔF/F [%]) aligned to lever extension with saline (black) and increasing doses of eticlopride: 0.1 mg/kg (pink), 0.25 mg/kg (green), 1.0 mg/kg (red), 2.5 mg/kg (orange), and 5.0 mg/kg (blue). N=5 mice. (**B**) Negative area under the curve (AUC) is reduced by eticlopride in a dose-dependent manner (RM ANOVA: $F_{(1.694, 6.777)}$=8.756, p=0.0150). (**C**) The rebound AUC is increased by eticlopride in a dose-dependent manner ($F_{(1.549, 6.197)}$=8.833, p=0.0181). (**D**) Total AUC is increased by eticlopride in a dose-dependent manner ($F_{(1.612, 6.448)}$=8.724, p=0.0170). (**E**) Dip duration is decreased by eticlopride in a dose-dependent manner ($F_{(1.392, 5.569)}$=36.37, p=0.0009). (**F**) The dip amplitude was not affected by eticlopride ($F_{(2.063, 8.251)}$=1.864, p=0.2147).

The online version of this article includes the following figure supplement(s) for figure 3:

**Figure supplement 1.** D2R antagonism increases cue-evoked DA release.

Individual CRF trials revealed varying durations of lever extension aligned ACh decreases that we sorted by lever press latency using a heatmap (***Figure 4A***). Based on this heatmap, we observed longer decreases associated with quick press latencies and two smaller decreases with slower press latencies with the second decrease co-occurring with the lever press. Thus, for press latencies <2 s the ACh decrease is a combination of a cue induced and movement associated pause. To separate the cue induced pause from the movement induced pause, we analyzed trials with press latencies >2 s. We still observed a decrease in the ACh dip duration with increasing doses of eticlopride (***Figure 4B***). Quantification of the negative AUC revealed a non-significant but trending decrease with increasing doses of eticlopride (***Figure 4C***), while the rebound AUC increased (***Figure 4D***), the total AUC increased (***Figure 4E***), and the dip duration (***Figure 4F***) decreased. Eticlopride had no effect on the ACh dip amplitude (***Figure 4G***). We also examined the effect of D2R antagonism on cue-induced DA release for trials with press latencies >2 s (***Figure 4—figure supplement 1***). Quantification of DA peak amplitude (***Figure 4—figure supplement 1B***) and AUC (***Figure 4—figure supplement 1C***)

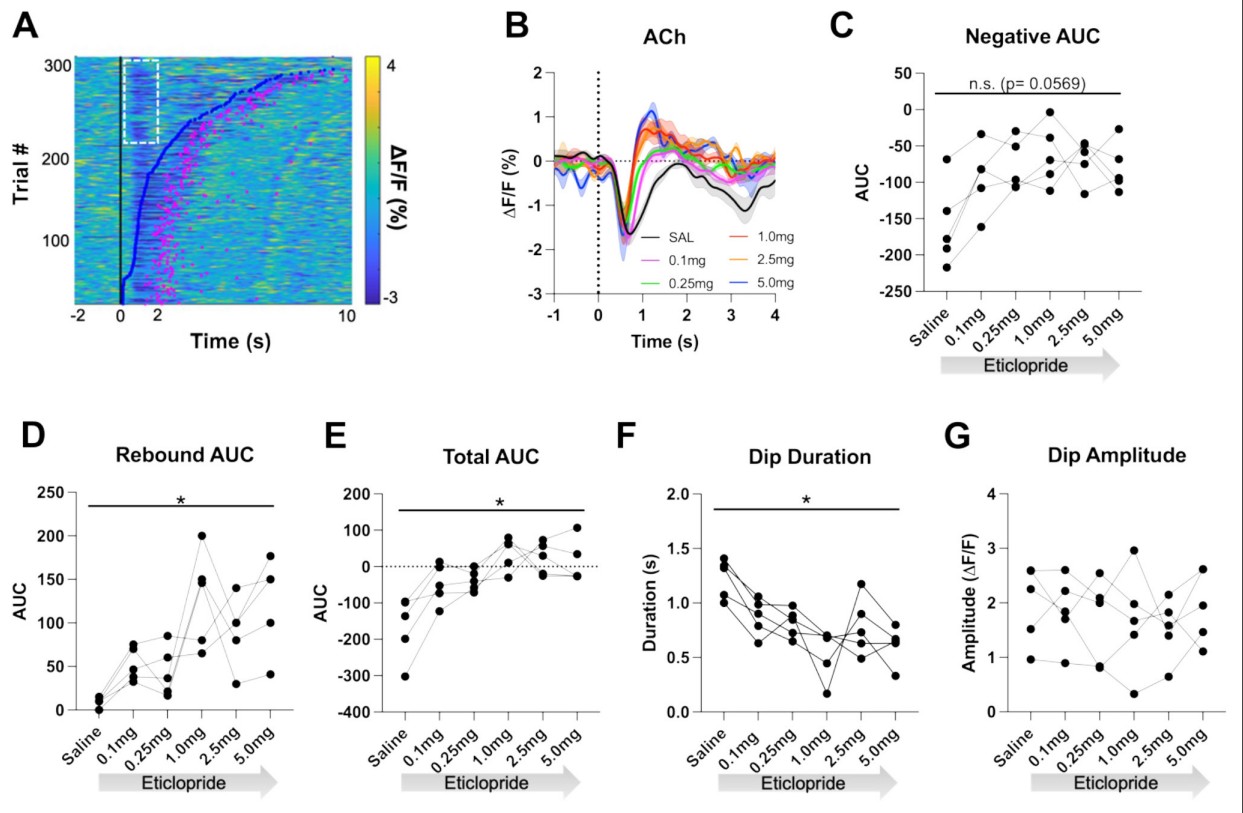

**Figure 4.** D2R antagonism shortens cue-evoked acetylcholine (ACh) dip and enhances rebound. (**A**) Heatmap of ACh responses aligned to lever extension (time = 0 s) for 300 individual trials (60 trials in 5 mice) and sorted by response length (bottom). Blue dots show the lever press, and the pink dots show the head entry for each trial. White dashed box represents the cue-evoked ACh response to the lever extension where press latencies are >2 s. N = 5 mice. (**B**) Changes in ACh fluorescence (ΔF/F [%]) aligned to lever extension for only trials with press latencies >2 s with increasing doses of eticlopride. (**C**) Negative area under the curve (AUC) is reduced by eticlopride in a dose-dependent manner (RM ANOVA: $F_{(2.237, 8.950)}$=3.911, p=0.0569). (**D**) Rebound AUC is enhanced by eticlopride in a dose-dependent manner ($F_{(1.667, 6.668)}$=8.143, p=0.0184). (**E**) Total AUC was increased by eticlopride in a dose-dependent manner ($F_{(1.597, 6.387)}$=8.542, p=0.0182). (**F**) Dip duration was significantly decreased by eticlopride in a dose-dependent manner ($F_{(1.657, 6.628)}$=6.729, p=0.0284). (**G**) Eticlopride had no effect on the dip amplitude ($F_{(2.722, 10.89)}$=0.5379, p=0.6503).

The online version of this article includes the following figure supplement(s) for figure 4:

**Figure supplement 1.** D2R antagonism enhances cue evoked DA release for trials with press latencies > 2s.

revealed an overall increase in both measures. Moreover, we found a significant increase between saline and 0.1 mg/kg eticlopride for DA peak amplitude (*Figure 4—figure supplement 1B*). Taken together, these results demonstrate that the cue-induced ACh decrease and rebound levels are regulated by cholinergic D2Rs.

## D2R blockade decreases negative and enhances positive correlations between DA and ACh

We further determined the relationship between ACh and DA levels within trials using a Pearson's correlation analysis. Using a lag analysis, we temporally shifted the ACh recording behind or in front of the DA recording to identify maximal points of correlation. During CRF trials, the strongest correlation is a negative correlation (*Figure 5A*, label 1, saline: Pearson's r=–0.475 ± 0.037, N=5) that occurs when ACh lags DA (Lag = –178.92 ± 14.38 ms), which accounts for 22% of the variance in the decrease in ACh being explained by the DA peak. This negative correlation, which reflects the decrease in ACh levels that follows the DA peak, is reduced with eticlopride in a dose-dependent manner (*Figure 5B*). Next, we found a small positive correlation (*Figure 5A*, label 2, saline: Pearson's r=0.039 ± 0.014) when ACh lags DA (Lag = –1.5 ± 0.138 s), which accounts for 0.15% of the variance in the ACh peak being explained by the DA peak. This positive correlation, which reflects the rebound in ACh, is significantly increased with eticlopride (*Figure 5C*).

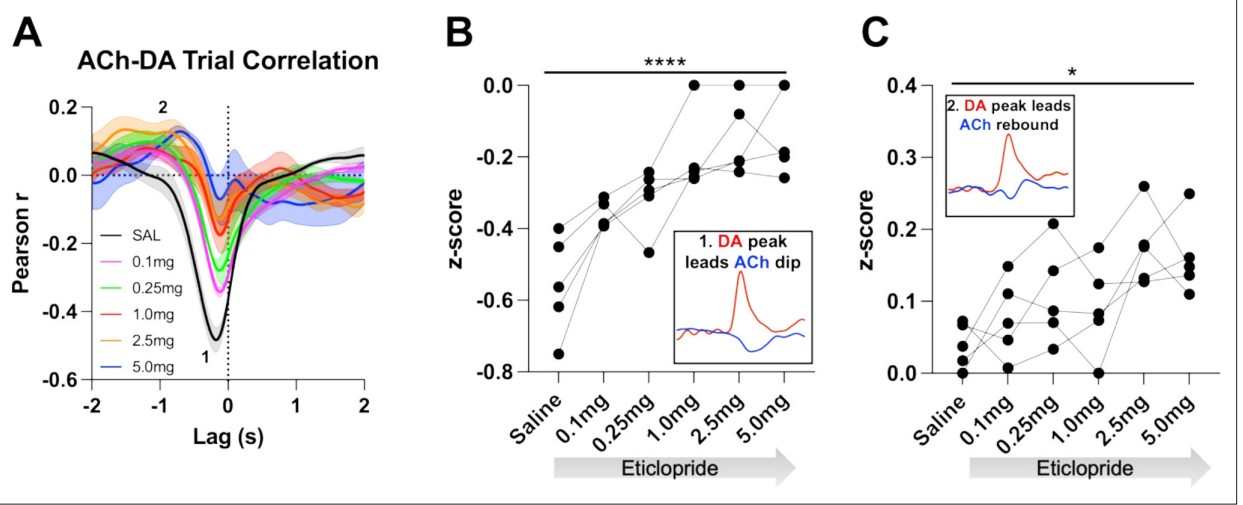

**Figure 5.** Task-dependent acetylcholine-dopamine (ACh-DA) interactions are altered by D2R antagonism at lever extension. (**A**) Correlation between ACh and DA during continuous reinforcement (CRF) trials with increasing doses of eticlopride in 5 C57BL/6J mice. The ACh signal moved in front of or behind the DA signal to identify points of highest correlation. The first correlation is a negative correlation (1) with ACh lagging DA (Lag = –178.92 ± 14.38 ms) and the second correlation is a positive correlation (2) with ACh lagging DA (Lag = –1.5 ± 0.138 s). N=5 mice. (**B**) The negative correlation with the DA peak leading the ACh dip (inset) is significantly reduced dose dependently by eticlopride (RM ANOVA: $F_{(2.596, 10.38)}$=18.67, p<0.0001). (**C**) The positive correlation with the DA peak leading the ACh rebound (inset) is enhanced by eticlopride in a dose-dependent manner ($F_{(2.326, 9.303)}$=4.694, p=0.0352).

We then analyzed these correlations during the ITI to determine whether they are only present during stimulus-induced DA/ACh signals or may represent a more general mechanism or coordination (***Figure 6A***). Of note, we looked for any interaction between DA and ACh regardless of event size. Like CRF trials, we observed two correlations during the ITI; DA peak leads ACh dip (Pearson's r=–0.355 ± 0.065 and Lag = –212.34 ± 16.91 ms) and DA peak leads ACh peak/rebound (Pearson's r=0.058 ± 0.021 and Lag = –1.41 ± 0.19 s), which accounts for 12.6% of the decrease in ACh being explained by the DA peak and 0.34% of the ACh peak by the DA peak, respectively. We found that eticlopride decreases the negative correlation in a dose-dependent manner (***Figure 6B***). Eticlopride

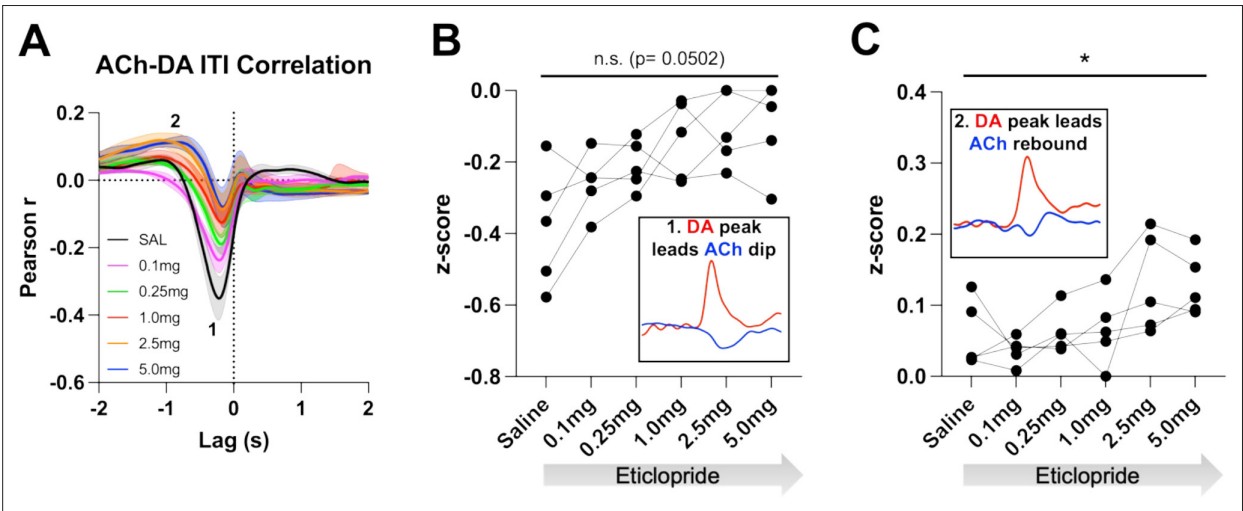

**Figure 6.** Acetylcholine-dopamine (ACh-DA) interactions are altered by D2R antagonism during the intertrial interval (ITI). (**A**) Correlation between ACh and DA during the ITI with increasing doses of eticlopride in C57BL/6J mice. We observe the same two correlations during the ITI: a negative correlation (1) with ACh lagging DA (Lag = –212.34 ± 16.91 ms) and a positive correlation (2) with ACh lagging DA (Lag = –1.41 ± 0.19 s). N = 5 mice. (**B**) The negative correlation with the DA peak leading the ACh dip (inset) is decreased by eticlopride in a dose-dependent manner (RM ANOVA: $F_{(1.850, 7.400)}$=4.689, p=0.0502). (**C**) The positive correlation with the DA peak leading the ACh rebound (inset) is increased dose dependently by eticlopride ($F_{(2.129, 8.515)}$=4.877, p=0.0373).

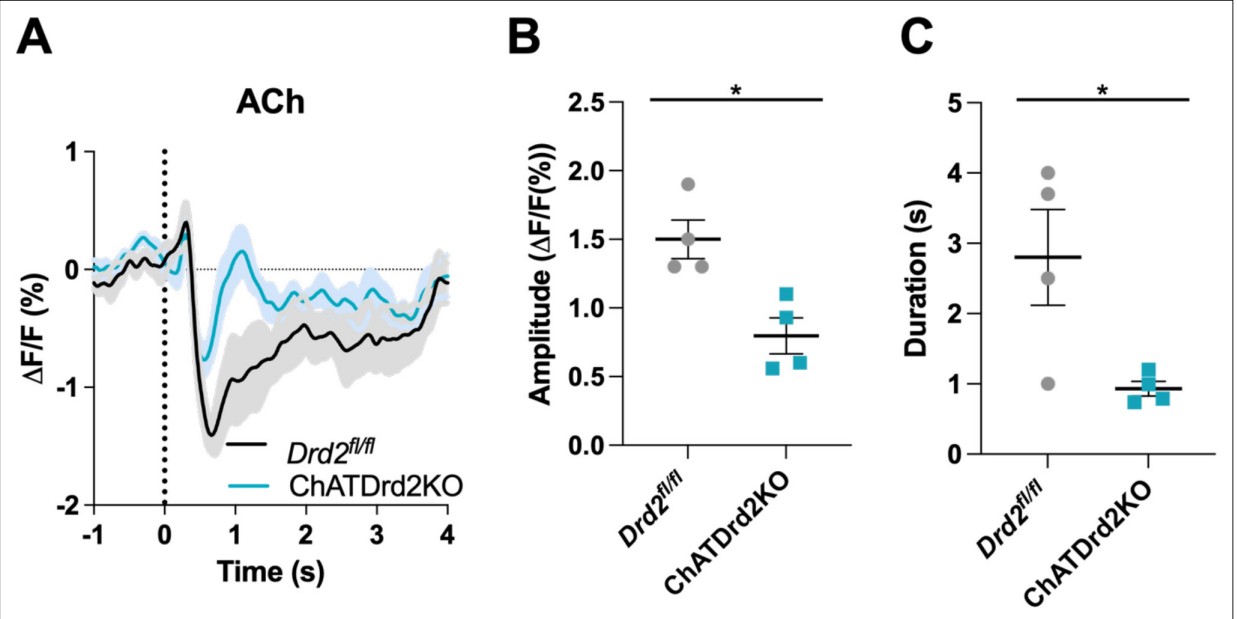

**Figure 7.** Selective D2R ablation from cholinergic interneurons (CINs) alters the cue-evoked acetylcholine (ACh) dip. (**A**) Changes in ACh fluorescence (ΔF/F [%]) aligned to lever extension for only trials with press latencies >2 s for *Drd2fl/fl* control (black) and ChATDrd2KO (blue) mice, N=4 mice/genotype. (**B**) Dip amplitude is significantly smaller in ChATDrd2KO animals compared to controls (t-test: p=0.0107). (**C**) Dip duration is significantly shorter in ChATDrd2KO mice compared to *Drd2fl/fl* controls (p=0.0351).

The online version of this article includes the following figure supplement(s) for figure 7:

**Figure supplement 1.** Selective D2R ablation from CINs does not alter cue-evoked DA release for trials with press latencies > 2 s.

also increased the positive correlation, which represents the ACh rebound (*Figure 6C*). These results indicate that DA-ACh correlations are dependent on D2Rs. While they are strong during salient cue presentations the relationship between both signals still exists during the ITI reflecting a general mechanism of co-regulation.

### Genetic inactivation of D2Rs from CINs shortens the decrease in ACh levels

Systemic eticlopride injections block all D2Rs. To determine the specific modulatory role that D2Rs present in CINs play in the cholinergic pause, we used mouse genetics to selectively inactivate D2Rs from CINs (ChATDrd2KO mice). We measured a smaller and shorter decrease in ACh levels in ChAT-Drd2KO mice compared to control mice in trials with press latencies >2 s (*Figure 7A–C*) or when taking all trials into account (data not shown). Note that the effects of D2R deletion differed from the highest dose of eticlopride in that ChATDrd2KO mice showed differences in the dip amplitude while eticlopride did not.

In contrast to ACh levels, stimulus-induced DA release was not altered in ChATDrd2KO mice (*Figure 7—figure supplement 1*). This result indicates that loss of cholinergic D2Rs does not affect stimulus-induced DA release and confirm that the effects of DA regulation of the ACh dip are mediated by CIN D2Rs and not an indirect effect by potential changes in DA levels.

### DA-mediated changes in ACh levels are dependent on CIN D2Rs

Next, we determined if D2Rs present in CINs are necessary for the effect of D2R antagonism on modulating the cue-induced changes in ACh levels. Control *Drd2fl/fl* mice were more sensitive to eticlopride than the C57BL/6J wild-type mice of *Figure 4* as they did not complete any trials with the two highest doses, 2.5 and 5.0 mg/kg (*Figure 8A–F*). Quantification of the ACh signal using the 3 lower doses revealed a decrease in the negative AUC (*Figure 8B*), an increase in the rebound AUC (*Figure 8C*), an increase in the total AUC (*Figure 8D*), and a decrease in dip duration (*Figure 8E*) that were comparable to what we measured in the C57BL/6J mice (*Figure 4*). Like the C57BL/6J mice, there was no effect

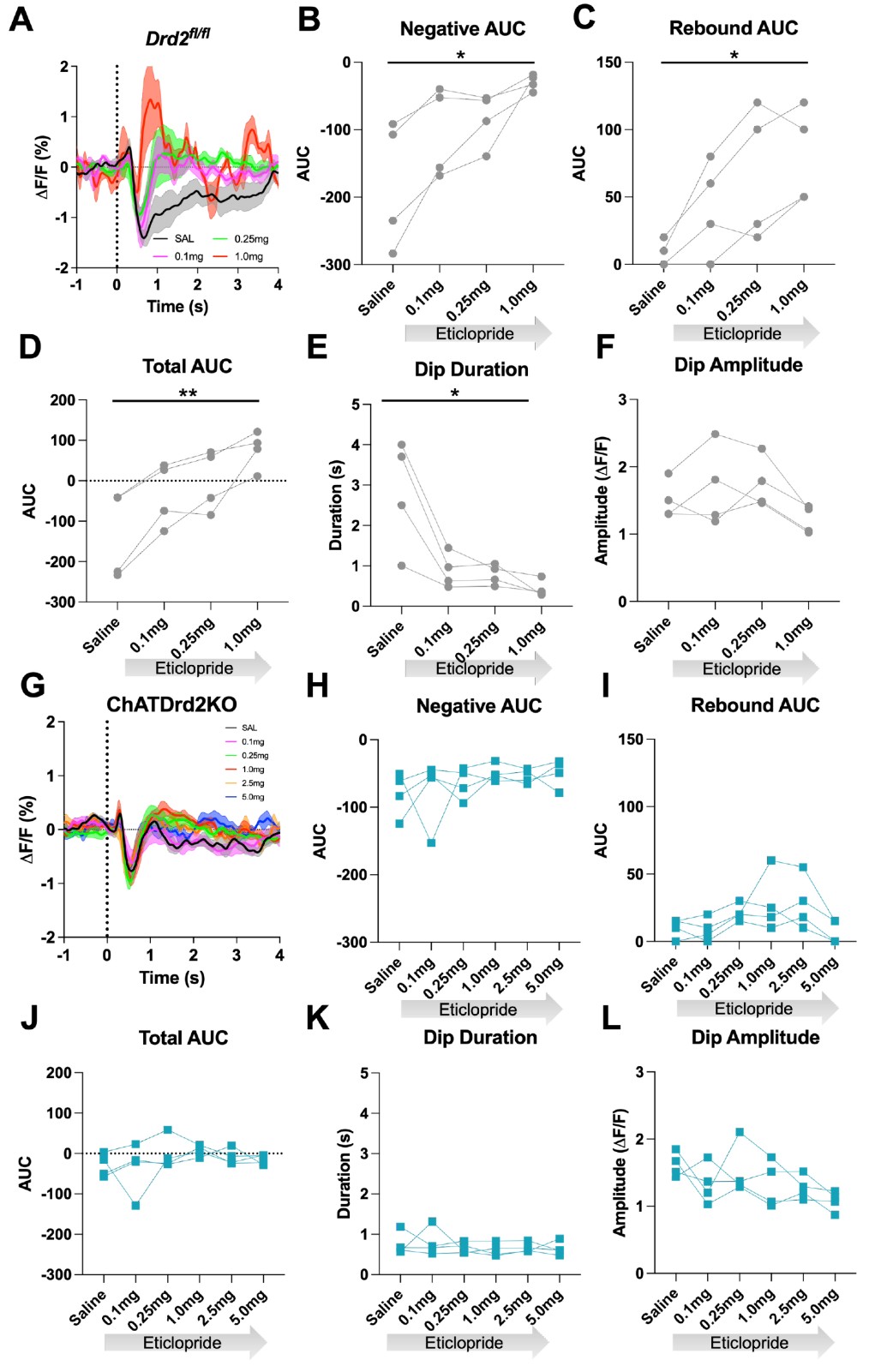

**Figure 8.** D2R antagonism does not alter the cue-evoked acetylcholine (ACh) dip in ChATDrd2KO mice. (**A**) Changes in ACh fluorescence (ΔF/F [%]) aligned to lever extension for only trials with press latencies >2 s for *Drd2^{fl/fl}* control mice with increasing doses of eticlopride. N = 4 mice (**B**) Negative area under the curve (AUC) is decreased by eticlopride in a dose-dependent manner (RM ANOVA: $F_{(1.387, 4.160)}$=8.541, p=0.0381). (**C**) Rebound AUC

*Figure 8 continued on next page*

*Figure 8 continued*

is increased by eticlopride in a dose-dependent manner ($F_{(1.642, 4.925)}$=10.21, p=0.0195). (**D**) Total AUC is increased dose dependently by eticlopride ($F_{(1.525, 4.676)}$=23.14, p=0.0047). (**E**) Dip duration is decreased by eticlopride in a dose-dependent manner ($F_{(1.664, 4.992)}$=9.279, p=0.0226). (**F**) Dip amplitude is not affected by eticlopride ($F_{(1.433, 4.300)}$=6.056, p=0.0606). (**G**) Changes in ACh fluorescence (ΔF/F [%]) aligned to lever extension for only trials with press latencies >2 s for ChATDrd2KO mice with increasing doses of eticlopride. N = 4 mice (**H**) Negative AUC is not affected by eticlopride ($F_{(1.663, 4.990)}$=0.7919, p=0.4803). (**I**) Rebound AUC is not affected by eticlopride ($F_{(1.706, 5.119)}$=2.857, p=0.1484). (**J**) Total AUC is not affected by eticlopride ($F_{(1.844, 5.532)}$=1.079, p=0.3958). (**K**) Dip duration is not affected by eticlopride ($F_{(1.848, 5.545)}$=0.4380, p=0.6516). (**L**) Dip amplitude is not affected by eticlopride ($F_{(2.073, 6.219)}$=2.546, p=0.1551).

on ACh dip amplitude with eticlopride (*Figure 8F*). In contrast, in ChATDrd2KO mice, we observed no effect of eticlopride on cue-induced changes in ACh levels (*Figure 8G*) neither on the negative AUC (*Figure 8H*), the rebound AUC (*Figure 8I*), total AUC (*Figure 8J*), dip duration (*Figure 8K*), or dip amplitude (*Figure 8L*). When *Drd2$^{fl/fl}$* and ChATDrd2KO mice were analyzed together we measured a gene × eticlopride interaction for negative AUC (genotype × dose: $F_{(3, 18)}$=3.113, p=0.0522), rebound AUC (genotype × dose: $F_{(3,18)}$=4.600, p=0.0147), total AUC (genotype × dose: $F_{(3,18)}$=8.106, p=0.0013), dip duration (genotype × dose: $F_{(3,18)}$=10.41, p=0.0003) but not dip amplitude (genotype × dose: $F_{(3,18)}$=1.611, p=0.2219). These results confirm that CIN D2Rs are responsible for the modulation of the ACh signal elicited by D2R antagonism.

## DA-mediated changes in DA-ACh correlations are dependent on CIN D2Rs

Next, we assessed the effect of CIN D2Rs in the ACh-DA co-regulation, again using Pearson's *r* correlation analysis and lag analysis. We found that the interaction between DA and ACh was greatly reduced (>twofold) in ChATDrd2KO mice compared to *Drd2$^{fl/fl}$* controls (*Figure 9*). The negative correlation with ACh lagging DA was significantly smaller in ChATDrd2KO mice during both CRF trials (*Figure 9A–B*) and the ITI (*Figure 9C–D*) compared to *Drd2$^{fl/fl}$* controls. We further examined the role of CIN D2Rs in the synchronization of DA and ACh activity using eticlopride to transiently block CIN D2Rs. *Drd2$^{fl/fl}$* control mice showed a strong negative correlation during CRF trials (*Figure 9A*: Pearson's *r*=–0.521 ± 0.038, N=4) with ACh lagging DA (Lag = –167.12 ± 18.82 ms), which accounted for 27% of variance of the decrease in ACh being explained by the DA peak. This relationship was reduced by eticlopride (*Figure 9—figure supplement 1B*). In addition, like in C57BL/6J mice we measured as a positive correlation (*Figure 9—figure supplement 1A*: Pearson's *r*=0.050 ± 0.030, N=4) with ACh lagging DA (Lag = –1.51 ± 0.138 s), which accounted for 0.25% of the variance of the ACh peak being explained by the DA peak. This relationship was increased with increasing doses of eticlopride (*Figure 9—figure supplement 1C*).

During the ITI, we measured a negative correlation between ACh and DA with ACh lagging DA (*Figure 9—figure supplement 2A*: Lag = –213.81 ± 34.14 ms) in *Drd2$^{fl/fl}$* control mice that was attenuated by eticlopride (*Figure 9—figure supplement 2B*), while the positive correlation with ACh lagging DA (Lag = –1.39 ± 0.208 s) was not significantly affected by eticlopride (*Figure 9—figure supplement 2C*). In ChATDrd2KO mice, at lever extension neither the negative correlation between ACh and DA (*Figure 9A*: Pearson's *r*=–0.231 ± 0.046, N=4) with ACh lagging after DA (Lag = –179.41 ± 30.66 ms, N=4) nor the positive correlation with ACh lagging DA (Lag = –1.635 ± 0.174 s) were modulated by eticlopride (*Figure 9—figure supplement 3*). During the ITI, we measured a negative correlation between ACh and DA (*Figure 9C*: Pearson's *r*=–0.153 ± 0.034, N=4; Lag = –282.62 ± 93.81 ms) that was significantly reduced in ChATDrd2KO mice compared to control mice (*Figure 9D*) and also was not affected by eticlopride (*Figure 9—figure supplement 4A-B*). These results confirm that D2Rs in CINs coordinate task-evoked and task-independent changes in ACh levels.

## D2R antagonism increases the latency to press in a CIN D2R-dependent manner

We then determined if manipulating CIN D2R function affects behavior in the CRF task. Since D2R blockade induces catalepsy (*Kharkwal et al., 2016*) we wondered whether Drd2 ablation or D2R antagonism alters behavioral responding (latency to press in the task), an indicator of motivated

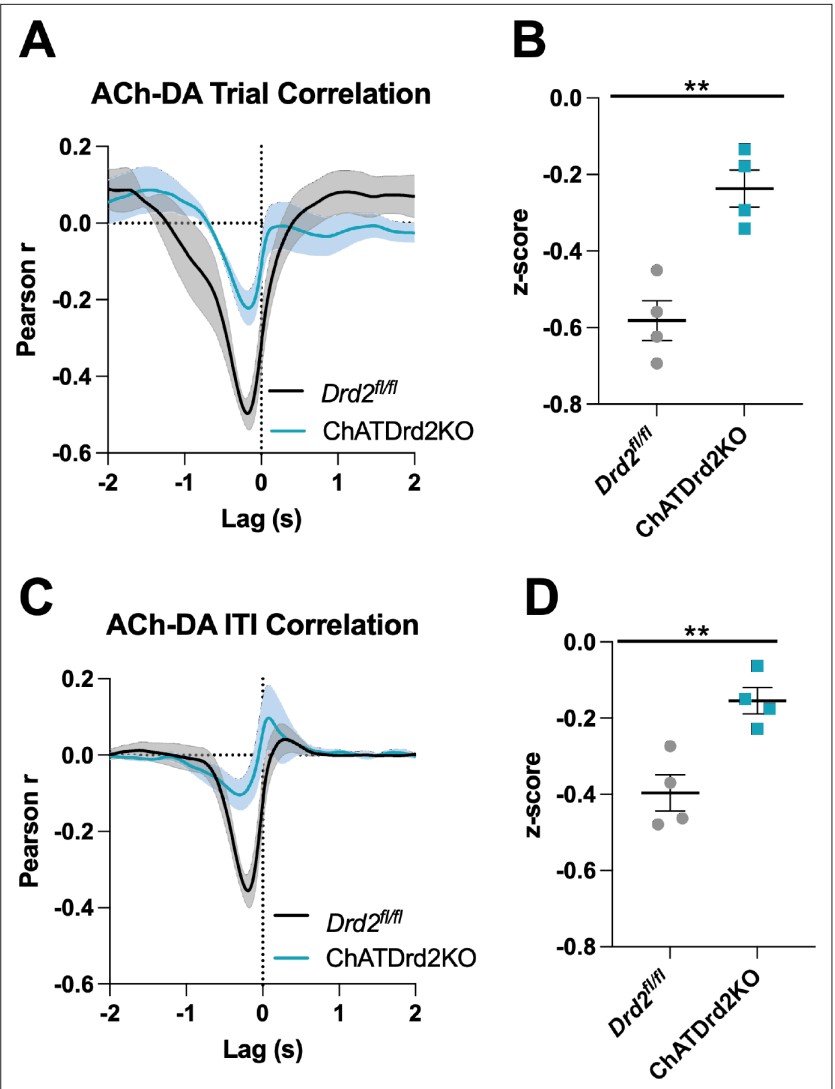

**Figure 9.** Acetylcholine-dopamine (ACh-DA) interactions are reduced in ChATDrd2KO mice. (**A**) Task-evoked correlation between ACh and DA for *Drd2fl/fl* control (black) and ChATDrd2KO (blue) mice. N = 4 mice/genotype (**B**) The negative correlation with ACh lagging DA is significantly reduced in ChATDrd2KO mice compared to *Drd2fl/fl* controls (t-test: p=0.0028). (**C**) Correlation between ACh and DA during the intertrial interval (ITI) for *Drd2fl/fl* control (black) and ChATDrd2KO (blue) mice. (**D**) The negative correlation of ACh lagging DA is significantly reduced in ChATDrd2KO mice compared to *Drd2fl/fl* controls (p=0.0062).

The online version of this article includes the following figure supplement(s) for figure 9:

**Figure supplement 1.** D2R antagonism alters cue-evoked ACh-DA interactions at the lever extension.

**Figure supplement 2.** D2R antagonism alters ACh-DA interactions in *Drd2fl/fl* control mice during the ITI.

**Figure supplement 3.** D2R antagonism does not alter cue evoked ACh-DA interactions in ChATDrd2KO mice at the lever extension.

**Figure supplement 4.** D2R antagonism does not alter general ACh-DA interactions in ChATDrd2KO mice during the ITI.

behavior. In C57BL/6J mice, we found that eticlopride significantly increased lever press latency in a dose-dependent manner (**Figure 10A**). Eticlopride had no effect on lever press latency in ChAT-Drd2KO mice, compared to *Drd2fl/fl* control mice (**Figure 10B**). Eticlopride also significantly increased the total number of complete trials, which was also observed in ChATDrd2KO mice but at higher doses than for *Drd2fl/fl* control mice (**Figure 10C**). The remaining effect in ChATDrd2KO mice is likely due to inhibition of D2Rs on indirect pathway neurons.

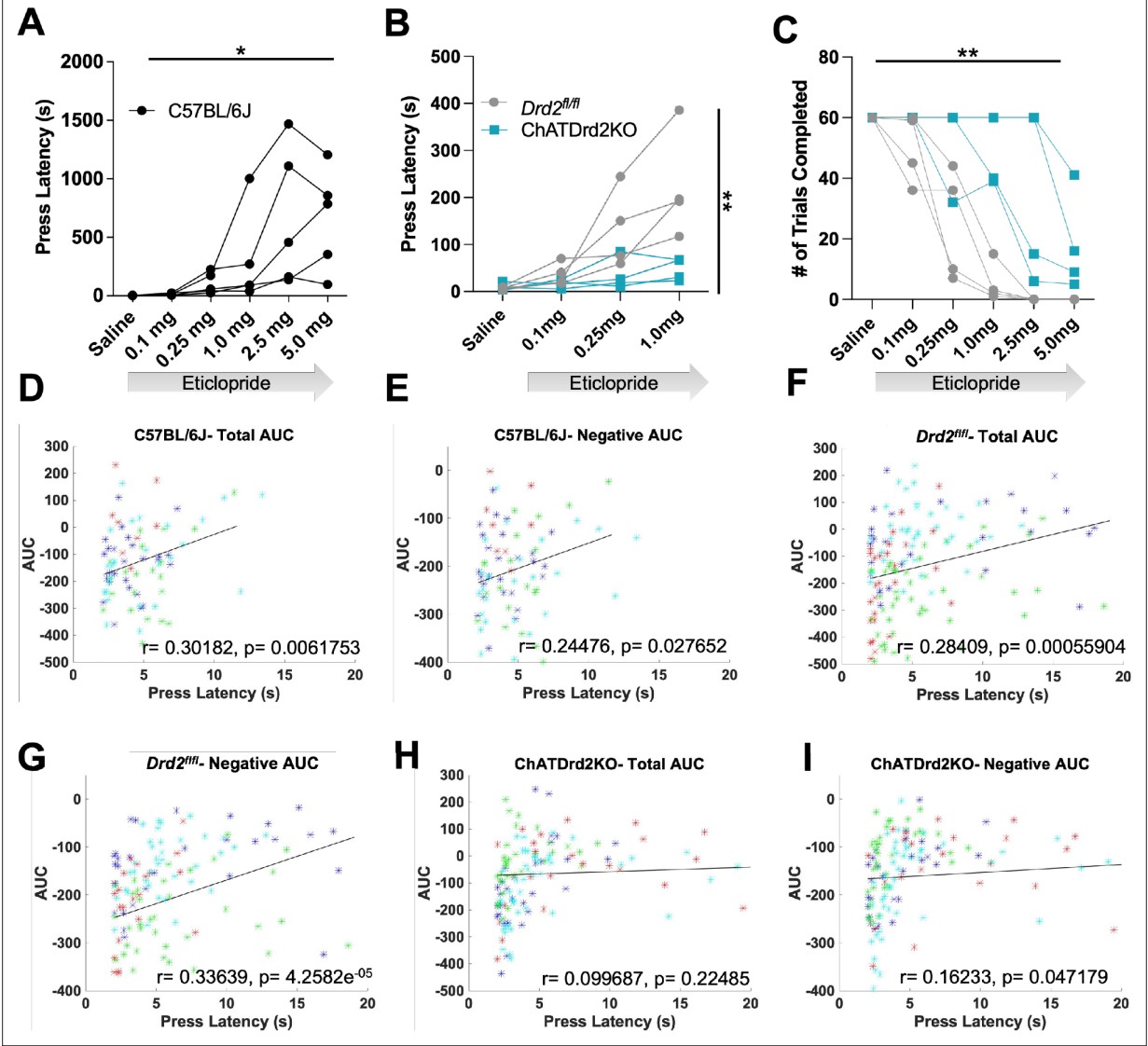

**Figure 10.** Behavioral responding correlates with acetylcholine (ACh) event size but is affected by D2R antagonism and ablation. (**A**) Lever press latency is increased by eticlopride in a dose-dependent manner in C57BL/6J mice (RM ANOVA: $F_{(1.383, 5.533)}$=6.369, p=0.0427). N = 5 mice (**B**) D2R antagonism does not increase lever press latency in ChATDrd2KO mice (blue squares) compared to $Drd2^{fl/fl}$ controls (gray circles) ($F_{(3,18)}$ = 5.664, p=0.0065, eticlopride × genotype). N = 4 mice/genotype (**C**) Eticlopride significantly increased the total # of trials completed in both ChatDrd2KO and $Drd2^{fl/fl}$ mice (genotype × dose effect: $F_{(5,30)}$ = 4.817), p=0.0024 (**D**) Total area under the curve (AUC) positively correlates with lever press latency in C57BL/6J mice ($r$=0.30182, p=0.0061753). (**E**) Negative AUC positively correlates with lever press latency in C57BL/6J mice ($r$=0.24476, p=0.027652). (**F**) Total AUC positively correlates with lever press latency in $Drd2^{fl/fl}$ control mice ($r$=0.28409, p=0.00055904). (**G**) Negative AUC positively correlates with lever press latency in $Drd2^{fl/fl}$ control mice ($r$=0.33639, p=4.2582e$^{-05}$). (**H**) Total AUC does not correlate with lever press latency in ChATDrd2KO mice ($r$=0.099687, p=0.22485). (**I**) Negative AUC positively correlates with lever press latency in ChATDrd2KO mice ($r$=0.16233, p=0.047179).

The online version of this article includes the following figure supplement(s) for figure 10:

**Figure supplement 1.** Press latency does not correlate with the size of the cue-evoked DA transient.

Next, we determined if the size of the stimulus-induced ACh decrease correlates with behavioral responding. To do this, we analyzed the correlation between the AUC and lever press latency for trials with press latencies >2 s to isolate the stimulus-induced ACh decrease from the lever press associated decrease. In C57BL/6J mice of *Figures 3–6*, we found a positive correlation between total AUC and press latency (*Figure 10D*). Similarly, in $Drd2^{fl/fl}$ control mice, we found a similar positive correlation between total AUC and press latency (*Figure 10F*). This correlation was driven by the decrease in ACh levels as the negative AUC correlated with press latency for C57BL/6J (*Figure 10E*) and $Drd2^{fl/fl}$ control (*Figure 10G*)

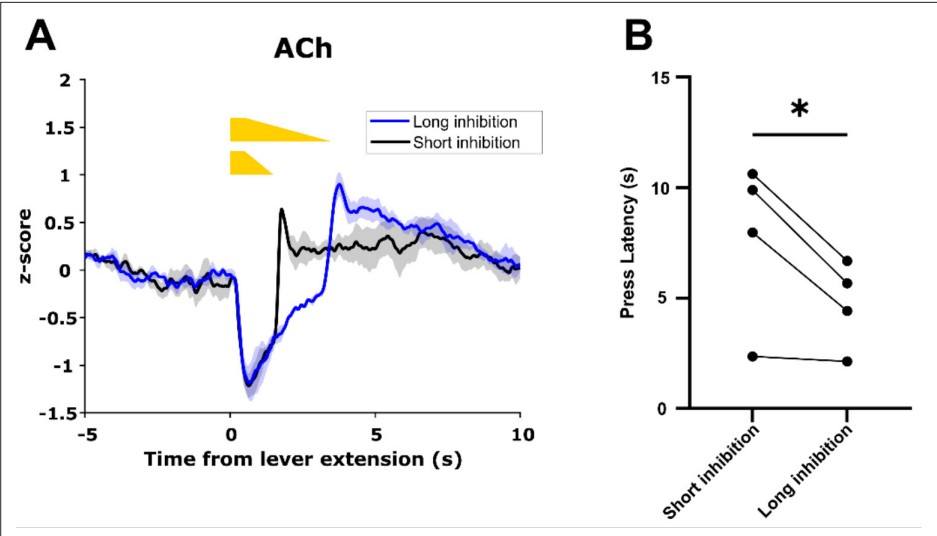

**Figure 11.** Acetylcholine (ACh) decrease duration regulates motivated behavior. (**A**) Short inhibition and long inhibition of cholinergic interneurons (CINs) during lever extension in the continuous reinforcement (CRF) task, respectively, produces short and long duration decreases in ACh (N=3 mice, one mouse excluded due to lack of GACh3.0 signal). (**B**) Mice exhibited longer lever press latency on short inhibition trials compared to long inhibition trials (N=4 mice, t-test: p=0.049).

mice. The correlation between total AUC and press latency was disrupted in ChATDrd2KO animals (*Figure 10H*), while there remained a weaker positive correlation between the negative AUC and press latency (*Figure 10I*). In addition, we analyzed the correlation between DA and behavioral responding and found no correlation between the DA AUC or max peak and lever press latency (*Figure 10—figure supplement 1*). This result suggests that cue-induced DA release does not regulate behavioral responding in the CRF task.

## ACh decrease duration regulates motivated behavior

To determine whether ACh decrease duration controls motivated behavior, we expressed GACh3.0 and Cre-dependent eNpHR3.0 in the dorsal medial striatum (DMS) of *ChAT*-IRES-Cre mice and trained these mice on the CRF task. After mice achieved criterion performance, we recorded GACh3.0 signals during the CRF task as above. On a subset of trials in each session, we inhibited CINs at lever extension in one hemisphere for a short (0.5 + 1 s ramp down) or long (0.5 + 3 s ramp down) duration to produce short and long duration decreases in ACh levels (*Figure 11A*). Consistent with the correlation we observed between ACh decrease size and lever press latency, we found lever press latency was longer during short inhibition trials compared to long inhibition trials (*Figure 11B*). These data are consistent with cue-induced changes in ACh levels regulating motivated behavior.

## Discussion

Here, we investigated the mechanism by which striatal DA regulates cue-induced changes in ACh levels during behavior. Understanding this mechanism is important because both neuromodulators coincidentally signal salient cues or outcomes during learning and motivated behavior and thus DA may regulate behavior via regulating ACh levels (*Apicella et al., 1992*). Moreover, it addresses the long-standing question of whether the ACh decrease is fully dependent on striatal DA.

By simultaneously recording task-evoked DA and ACh levels in mice we made several observations: first, we observed that changes in striatal DA and ACh levels are induced by reward-predicting stimuli and the time locked signals develop in parallel with learning. Second, we found that pharmacological and genetic inactivation of D2Rs does not completely abolish the stimulus-induced decrease in ACh, but it does shorten the decrease and enhances rebound levels. Third, using correlational analysis, we found a relationship between DA and ACh that was strongest in response to lever extension as a reward-predicting cue but still present during the ITI. This relationship was disrupted by

D2R inactivation. Fourth, we found that D2R antagonism increased the latency to lever press during behavior, but this was abolished when we inactivated CIN D2Rs. The size of the cue-evoked decrease in ACh levels correlated with lever press latency. and artificial CIN-inhibition at cue presentation influenced press latency, with longer inhibition leading to shorter press latencies. Altogether, these findings indicate that DA and cholinergic D2Rs are necessary for controlling the shape of the ACh signal and the coordinated activity between DA and ACh. Moreover, the cue-induced changes in ACh levels may have motivational significance by regulating behavioral responding.

## Cue-induced changes in striatal DA and ACh levels are time locked and develop in parallel with learning

The changes in DA and ACh levels that we recorded in the Pavlovian conditioning task are consistent with DA neurons and CINs encoding unexpected rewards and reward-predicting cues (*Aosaki et al., 1994b*; *Joshua et al., 2008*; *Morris et al., 2004*; *Schultz et al., 1997*; *Watanabe and Kimura, 1998*). Like these previous studies, that assayed neuronal activity, we see a robust increase in DA levels and a decrease in ACh levels to unexpected reward that diminish as the reward becomes expected. These data show that the neurotransmitter levels of both, DA and ACh, follow neuronal activity of their respective neurons with a sub-second kinetics. The fast induction of the ACh decrease is particularly striking as it suggests fast degradation or diffusion of ACh.

In addition, we observed similar changes in DA and ACh levels to the conditioned stimulus and not the unconditioned stimulus, which occur in parallel over learning. This is consistent with previous studies showing that DA neurons and CINs respond to salient and conditioned stimuli. Moreover, we found that these changes in DA and ACh levels correlate with behavioral responding. However, this correlation was not observed in all tested mice due to the nature of the Pavlovian task. In the Pavlovian task, animals have the possibility to learn an association between the CS+ and reward. Consequently, some animals may show anticipatory responding during the CS+ (head entries into the reward port). However, because anticipatory behavior is not required to obtain a reward, some of the tested mice did not exhibit anticipatory head poking. Strikingly, the development of a DA and ACh signal over time indicate that these animals are nevertheless learning the stimulus-reward association. In conclusion, these data show that co-incident task-evoked changes in ACh and DA levels in mice follow what has been described at the level of neuronal activity level in primates (*Joshua et al., 2008*; *Morris et al., 2004*; *Schultz et al., 1997*).

## D2R inactivation in CINs shortens but does not abolish the cue- induced decrease in ACh levels

Salient and conditioned stimuli are known to induce pauses in CIN and TAN firing in rodents and primates; however, the dependence on DA and CIN D2Rs for pause induction is widely debated (*Aosaki et al., 1994b*; *Morris et al., 2004*; *Watanabe and Kimura, 1998*; *Zhang and Cragg, 2017*). Thus, to determine the role D2Rs play in modulating the stimulus induced ACh decrease, we pharmacologically blocked D2Rs or selectively ablated Drd2 from CINs and measured ACh and DA levels in the CRF task. We found that D2R blockade or ablation shortened the decrease in ACh levels, which we quantified by calculating the dip duration. Moreover, in our control mice, D2R blockade also decreased the negative and increased total and rebound AUCs in a dose-dependent manner while the dip amplitude was unaffected. In contrast, D2R blockade had no effect on the stimulus-induced ACh signal in ChATDrd2KO mice. These data reveal that the initiation of the decrease in ACh levels is not dependent on CIN D2Rs. Instead, cholinergic D2Rs are important for modulating the duration of the stimulus-induced ACh decrease. Our data provide clarity on the controversial role that DA plays in the regulation of the ACh decrease and suggest that the stimulus-induced ACh decrease in vivo is not entirely DA or D2R-dependent as studies in primates and slice physiology studies have suggested (*Aosaki et al., 1994b*; *Ding et al., 2010*; *Watanabe and Kimura, 1998*). Our data further indicates that slice physiology studies in rodents where optogenetic stimulation of DA terminals or caged DA induced CIN pauses are abolished by D2R antagonists or CIN-selective ChATDrd2KO mice are not fully capturing the physiology underlying the natural pause (*Augustin et al., 2018*; *Chuhma et al., 2014*; *Kharkwal et al., 2016*; *Straub et al., 2014*; *Wieland et al., 2014*).

## DA levels are unaffected in ChATDrd2KO mice

Slice and in vivo stimulation studies have shown that stimulation of ACh release induces DA release via activation of nicotinic receptors on DA terminals (*Cachope and Cheer, 2014*; *Cachope et al., 2012*; *Sulzer et al., 2016*; *Threlfell et al., 2012*). Thus, it is surprising that despite a change in cue-induced ACh signal the cue-induced DA signal is largely unchanged in ChATDrd2KO mice. Especially, we hypothesized that the rebound in ACh levels observed in the ChATDrd2KO mice may enhance or prolong the DA peak, but this was not observed. Moreover, although eticlopride enhanced DA release, this enhancement was measured in wild-type (*Figure 3—figure supplement 1*) and ChATDrd2KO mice (data not shown). Since ChATDrd2KO mice do not show the eticlopride-induced rebound in DA levels, we believe that the enhanced initial peak in the eticlopride condition is due to antagonism of D2 auto-receptors on DA terminals. It is unclear why there is no additional effect on DA release as a consequence of the ACh rebound but it could be that under the behavioral conditions of the CRF task nicotinic receptors on DA neurons are desensitized.

## DA and ACh correlation during task-dependent behaviors

Our approach to simultaneously image both DA and ACh in the same animal allowed us to examine the relationship between these two neuromodulators within trials. In both C57BL/6J and *Drd2^{fl/fl}* control mice, we identified a strong negative correlation with the cue-induced DA peak leading the ACh dip that is attenuated by D2R antagonism. This strong negative correlation between DA and ACh is significantly reduced in ChATDrd2KO mice compared to *Drd2^{fl/fl}* controls and ChATDrd2KO mice are unaffected by D2R blockade. We also found a weaker positive correlation with DA leading ACh that is enhanced by D2R antagonism. We believe that this positive correlation reveals the rebound in ACh activity following the decrease that is blunted by D2R activation at baseline. These data suggest, as discussed above, that CIN D2Rs not only modulate the ACh pause but also the rebound activity.

Electrical stimulation in brain slices revealed on/off kinetics for dLight1.2 of 9.5 and 90 ms, respectively (*Labouesse and Patriarchi, 2021*). In contrast, the kinetics for the ACh3.0 sensor after electrical stimulation were slower with on/off kinetics of 90–105 ms and 0.9–3.7 s, respectively (*Jing et al., 2020*). Here, we measured a decrease in striatal ACh levels in vivo with an onset of 206 ms after optogenetic inhibition of CINs. This is significantly shorter than the reported on/off kinetics for the ACh3.0 sensor in brain slices and may be due to a faster clerance of ACh levels in vivo. Nevertheless, it is possible that the lag of the cue-induced decrease in ACh in relationship to the peak in DA is due to differences in the on/off kinetics of the two sensors and not a natural reflection of when both events occur.

## Implications for behavior

In addition to the effects on ACh decrease duration, we found that CIN D2Rs also regulate the magnitude of ACh rebound levels, acting as a mechanism to constrain ACh rebounds after the decrease. Currently, it is unknown which role the rebound in ACh plays during behavior. Generally, CINs are thought to inhibit spiny projections neurons (SPNs) via nicotinic activation of local interneurons or via muscarinic $M_2/M_4$-mediated inhibition of corticostriatal inputs (*English et al., 2012*; *Faust et al., 2015*; *Pakhotin and Bracci, 2007*; *Witten et al., 2010*). Thus, a larger decrease may lead to disinhibition and higher rebound to a stronger inhibition of SPNs. The D2R antagonism decreases the first and enhances the second which may inhibit movement initiation leading to the longer latency in lever pressing. Consistent with this, we observed that the size of the cue-induced ACh decrease correlates with press latency (the larger the decrease the shorter the latency). Surprisingly, this relationship also holds true for lever presses that were performed long after the cue-induced ACh signal reverted to normal. To address whether the duration of cue-induced decrease in ACh levels alter press latencies, we used optogenetic tools to artificially induce shorter and longer decreases in ACh at cue presentation. We found that short inhibition of CINs increased press latency compared to long inhibition of CINs. However, the longer inhibition also led to a larger rebound and both changes may be responsible for the change in press latency. Nevertheless, this suggests that the cue-evoked change in ACh levels regulate the motivation to initiate the lever press. This finding is consistent with recent inhibition studies in which CIN inhibition in the NAc during Pavlovian to instrumental transfer enhanced the ability of the Pavlovian cue to invigorate behavior (*Collins et al., 2019*).

What does an altered ACh signal mean for learning? CIN-selective D2R knock out mice learn the Pavlovian task presented in *Figure 1* as well as control littermates (data not shown). This suggests that even with an altered ACh signal mice still can learn cue-reward associations. Similarly, we recently described that enhancing the cue-induced decrease in ACh levels by selective overexpression of D2Rs in CINs of the NAc (D2R-OE$_{NAcChAT}$ mice) did not affect Pavlovian learning but was associated with a deficit in Go/NoGo learning (*Gallo et al., 2022*). This is consistent with prior studies reporting that striatal ACh is not necessary for initial learning but is important for behavioral performance when task contingencies change (*Aoki et al., 2015*; *Bradfield et al., 2013*; *Brown et al., 2010*; *Favier et al., 2020*; *Okada et al., 2014*; *Okada et al., 2017*; *Ragozzino et al., 2009*). Alternatively, the changes in the ACh signal observed in D2R overexpressing mice are responsible for suppressing the inhibitory learning that is required for the acquisition of the NoGo behavior.

Surprisingly, we did not measure an increase in press latencies in D2R KO mice, despite the short-ened dip duration. This discrepancy may be due to compensation in the regulation of press latencies via other mechanisms that have developed during development or as the consequence of the chronic change in D2R levels. Such compensatory mechanisms may not have the time to develop in response to the acute action of the antagonist. Alternatively, the correlation between cue-induced dip duration and press latency could be an epiphenomenon. For example, D2R antagonism may change both, tonic ACh levels (*Ikarashi et al., 1997*) and cue-induced changes in ACh, but it is only the first one that regulates press latencies. The optogenetic inhibition experiment argues otherwise as the short inhibition of CINs increased press latency in comparison to long inhibition of CINs, suggesting that differences in phasic ACh levels are sufficient to regulate motivated behavior. Further investigation is needed to determine whether compensatory mechanisms overcome changes in cue-induced changes in ACh levels in D2R KO mice to normalize motivated behavior.

In conclusion, our data demonstrate that the stimulus-induced ACh signal is multiphasic, encompassing a DA-dependent and a non-DA-dependent component. Striatal DA is responsible for confining the temporal boundaries of the ACh signal and preventing rebound excitation via CIN D2R. Notably, we also find a positive correlation between the size of the stimulus-induced decrease in ACh levels and behavioral responding, which implicates a role for the cue-induced ACh signal in the motivation to initiate actions. Further dissection of ACh DA co-regulation in the striatum will be essential for a better understanding of how both signals regulate behavior.

## Materials and methods

**Key resources table**

| Reagent type (species) or resource | Designation | Source or reference | Identifiers | Additional information |
|---|---|---|---|---|
| Gene (*Mus musculus*) | C57BL/6J | JAX | Cat. #: 000664 | |
| Gene (*Mus musculus*) | *ChAT*-IRES-Cre | *Rossi et al., 2011* | JAX stock Cat. #: 031661 | |
| Gene (*Mus musculus*) | *Drd2$^{fl/fl}$* | *Bello et al., 2011* | | |
| Genetic reagent (adeno-associated virus) | AA5-hSYN-dLight1.2 | Addgene | Cat. #: 111,068 | |
| Genetic reagent (adeno-associated virus) | AA5-hSYN-ACh3.0 | Addgene | Cat. #: 121,922 | |
| Genetic reagent (adeno-associated virus) | AAV5-EF1α-DIO-eNpHR3.0 | UNC Vector Core | Cat. #: 129,381 | |
| Chemical compound, drug | Eticlopride | Tocris | Cat. #: 1847 | 0.1, 0.25, 1.0, 2.5, 5.0 mg/kg |
| Chemical compound, drug | Scopolamine | Tocris | Cat. #: 1414 | 15 mg/kg |
| Software, algorithm | Analysis | Synapse | Tucker-Davis Technologies | |
| Software, algorithm | Analysis | MATLAB | MathWorks | DIO Link (Pavlovian): 10.6084 /m9.figshare.19586026 DIO Link (CRF): 0.6084 /m9.figshare.19586032 |
| Software, algorithm | Analysis | Prism 9 | GraphPad | |

## Animals

Adult male and female C57BL/6J (JAX stock #000664) mice (*Figures 1–6 and 10*) were bred in house. For control and KO animals (*Figures 7–10*): double-transgenic mice were generated by crossing heterozygous *ChAT*-IRES-Cre mice (*Rossi et al., 2011*) (JAX stock #031661) to homozygous *Drd2*fl/fl (*Drd2*fl/fl) mice (*Bello et al., 2011*). Control (*Drd2*fl/fl) and ChATDrd2KO (*ChAT*-IRES-Cre x *Drd2*fl/fl) mice are littermates, bred in house and back crossed onto C57BL/6J background. Mice were housed 1–4 per cage for most experiments on a 12 hr light/dark cycle, and all experiments were conducted in the light cycle. All experimental procedures were conducted following NIH guidelines and were approved by Institutional Animal Care and Use Committees by Columbia University and the New York State Psychiatric Institute (NYSPI protocol #1621).

## Pharmacology

Intraperitoneal injections of saline, eticlopride (Tocris Cat. No. 1847) (0.1, 0.25, 1.0, 2.5, and 5.0 mg/kg) or scopolamine (Tocris Cat. No. 1414) (15 mg/kg) were administered 1 hr before behavioral testing. To generate a dose-response curve with eticlopride, saline days alternated with drug days and the drug was administered in order from the lowest to highest dose.

## Surgical procedures

Mice (≥8 weeks old) were induced with 4% isoflurane and maintained at 1–2% throughout the procedure. Mice were bilaterally injected with 450 nL/hemisphere with either AA5-hSYN-dLight1.2 (Addgene) (*Patriarchi et al., 2018*) or AA5-hSYN-ACh3.0 (Addgene) (*Jing et al., 2020*) (also known as GRAB-ACh3.0) into separate hemispheres of the DMS using stereotactic Bregma-based coordinates: AP, +1.1 mm; ML, ±1.4 mm; DV, –3.1 , –3.0 , and –2.9 mm (150 nL/DV site). The on/off kinetics for the dLight1.2 sensor are 9.5 and 90 ms, respectively (*Labouesse and Patriarchi, 2021*). The on/off kinetics for the ACh3.0 sensor are 90–105 ms and 0.9–3.7 s, respectively (*Jing et al., 2020*). For the optogenetic inhibition experiment, mice were co-injected unilaterally with AA5-hSYN-ACh3.0 and AAV5-EF1α-DIO-eNpHR3.0 (UNC Vector Core) into the DMS. Following virus injection, 400 μm fiber optic cannulas (Doric, Quebec, Canada) were carefully lowered to a depth of –3.0 mm and fixed in place to the skull with dental cement anchored to machine mini-screws. Groups of mice used for experiments were housed in a counterbalanced fashion that accounted for sex, age, and home cage origin. Cannula-implanted mice began behavioral training 4 weeks after surgery. At the end of experiments, animals were perfused, and brains were processed post hoc to validate virus expression and optic fiber location as in *Gallo et al., 2022*.

## In vivo fiber photometry and optogenetics

Fiber photometry equipment was set up using two 4-channel LED driver (Doric) connected to two sets of a 405 and a 465 nm LEDs (Doric, cLED_405, and cLED_465). The 405 nm LEDs were passed through 405–410 nm bandpass filters, while the 465 nm LEDs were passed through 460–490 nm GFP excitation filters using two 6-port Doric minicubes. A 405 and 465 LEDs were then coupled to a dichroic mirror to split excitation and emission lights. Two low-autofluorescence patch cords (400 μm/0.48 NA, Doric) arising from the two minicubes were attached to the cannulas on the mouse's head and used to collect fluorescence emissions. These signals were filtered through 500–540 nm GFP emission filters via the same minicubes coupled to photodetectors (Doric, gain set to DC low). Signals were sinusoidally modulated, using Synapse software and RZ5P Multi I/O Processors (Tucker-Davis Technologies), at 210 and 330 Hz (405 and 465 nm, respectively) to allow for low-pass filtering at 3 Hz via a lock-in amplification detector. 405 and 465 nm power at the patch cord were set to 30 μW or below. For acute optogenetic inhibition via eNpHR3.0, amber light (595 nm LED, Doric) was applied through the same optic fiber using a short and long optogenetic protocol: (i) 500 ms square pulses at 1 mW + 1 s ramp down and (ii) 500 ms + 3 s ramp down. The 595 nm light was passed through a 580–680 F2 port (photodetector removed) of the same 6-port minicube (*Pisansky et al., 2019*). Optogenetic inhibition was performed in the home cage or during lever extension in a subset of trials in the CRF task.

## Photometry data processing

All photometry and behavioral data utilized custom in-house MATLAB analysis scripts. Photometry signals were analyzed as time-locked events aligned to the lever extension (CRF) or tone onset

(Pavlovian) of each trial. The 405 nm channel was used to control for potential noise/movement artifacts and the 465 nm channel was used to detect the conformational modulation of either the GACh3.0 sensor by ACh or the dLight1.2 sensor by DA. Both demodulated signals were extracted as a 15 s window surrounding the event, which was denoted as time = 0. Both the signals were down sampled by a factor of 10 using a moving window mean. The change in fluorescence, ΔF/F (%), was defined as (F-F0)/F0 × 100, where F represents the fluorescent signal (465 nm) at each time point. F0 was calculated by applying a least-squares linear fit to the 405 nm signal to align with the 465 nm signal (*Calipari et al., 2016*). To normalize signals across animals and sessions, we calculated a local baseline fluorescence value for each trial using the average of the 5 s period preceding the event and subtracted that from the signal. The daily average GACh3.0 and dLight1.2 traces were calculated using session average traces from individual mice. ACh dip and DA peak amplitudes were calculated as the maximal change of the signal that was at least 1 or 2 STD below or above the local baseline, respectively. ACh dip duration was calculated using the last and the first zero crossings preceding and following the decrease in ACh levels. Total AUC was calculated as the area of all the three components of the ACh signal (initial peak, decrease, and rebound). Negative AUC was calculated as the area for only the negative component. Rebound AUC was calculated as the area of the positive component immediately following the decrease in ACh levels. The AUC analysis was restricted to a 5 s time window following the task event. Individual CRF trial (ΔF/F [%]) traces were used for correlation analysis for CRF trials. For the ITI correlation, we examined any interaction between dLight1.2 and GACh3.0 regardless of event size during the variable 40 s of the ITI.

## Operant apparatus

Four operant chambers (model Env-307w; Med-Associates, St. Albans, VT) equipped with liquid dippers were used. Each chamber was in a light- and sound-attenuating cabinet equipped with an exhaust fan, which provided 72 dB background white noise in the chamber. The dimensions of the experimental chamber interior were 22×18 × 13 cm, with flooring consisting of metal rods placed 0.87 cm apart. A feeder trough was centered on one wall of the chamber. An infrared photocell detector was used to record head entries into the trough. Raising of the dipper inside the trough delivered a drop of evaporated milk reward. A retractable lever was mounted on the same wall as the feeder trough, 5 cm away. A house light located on the wall opposite to trough illuminated the chamber throughout all sessions.

## Dipper training

Four weeks after surgery, mice underwent operant training. Mice were weighed daily and food restricted to 85–90% of baseline weight; water was available ad libitum. In the first training session, 20 dipper presentations were separated by a variable ITI and ended after 20 rewards were earned or after 30 min had elapsed, whichever occurred first. Criterion consisted of the mouse making head entries during 20 dipper presentations in one session. In the second training session, criterion was achieved when mice made head entries during 30 of 30 dipper presentations.

## Pavlovian conditioning

Mice were trained for 16 consecutive days in a Pavlovian conditioning paradigm, which consisted of 12 conditioned stimulus-positive (CS+) trials and 12 unconditioned stimulus-negative (CS−) trials occurring in a pseudorandom order. Each trial consisted of an 80 dB auditory cue presentation for 10 s of an 8 kHz tone or white noise (counterbalanced between mice) and after cue offset a milk reward was delivered only in CS+ trials, whereas no reward was delivered in CS− trials. There was a 100 s variable ITI, drawn from an exponential distribution of times. Head entries in the food port were recorded throughout the session, and anticipatory head entries during the presentation of the cue were considered the conditioned response. Anticipatory responding was calculated as the difference in nose poking during the CS+ quintile with the maximum response (Q4 or 5) and the first quintile.

## CRF schedule

For lever press training, lever presses were reinforced on a CRF schedule. Levers were retracted after each reinforcer and were presented again after a variable ITI (average 40 s). The reward consisted

of raising the dipper for 5 s. The session ended when the mouse earned 60 reinforcements, or 1 hr elapsed, whichever occurred first. Sessions were repeated daily until mice achieved 60 reinforcements.

### Data analysis

To determine the sample size for detecting whether D2R antagonism alters the cue-evoked change in ACh levels we performed a power analysis on the first cohort of wild-type mice using total AUC from *Figure 4* as the outcome measure. The analysis was performed in Gpower using the repeated measures, within factors ANOVA power analysis. In the first step, we calculated a partial eta square using F*df_effect/(F*df_effect + df_error) which then was used to calculate the effects size f. A non-sphericity correction epsilon was calculated in Prism GraphPad to quantify how variances differ between different repeated measures. A correlation matrix between repeated measures was used to determine the average correlation between repeated measures. Based on this we calculated for AUC (eta square: 0.681, effect size f: 1.46, alpha: 0.05, power: 0.8, repeated measures: 6, corr among repeated measures: –0.068, and non-sphericity correction: 0.3193) a sample size of n=5. A similar analysis for press latency (input: eta square: 0.620, effect size f: 1.28, alpha: 0.05, power: 0.8, repeated measures: 6, corr among repeated measures: 0.1294, and non-sphericity correction: 0.278) revealed a sample size of n=5. Expecting similar effect sizes in follow up experiments we aimed for a sample size of 5 mice per group when analyzing the CIN-selective D2R KO mice and their littermate controls. Statistical analyses were performed using GraphPad Prism 9 (GraphPad), MATLAB (MathWorks). Data are generally expressed as mean ± SEM. Paired and unpaired two-tailed Student's t-tests were used to compare 2-group data, as appropriate. Multiple comparisons were evaluated by one- or two-way ANOVA and Bonferroni's post hoc test, when appropriate. In rare cases of values missing in repeated measures samples, the data were analyzed by fitting a mixed effects model, as implemented by Prism 9. Photometry correlation analyses were performed using Pearson's correlation coefficients. We used a variance explained statistical analysis ($R^2$) to determine the % of variance in our correlation analyses (e.g. a correlation of 0.5 means $0.5^2 \times 100 = 25\%$ of the variance in Y is 'explained' or predicted by the X variable). When comparing correlation values, Fisher's transformation was used to convert Pearson correlation coefficients to z-scores. A p-value of <0.05 was considered statistically significant. Both male and female mice were analyzed; however, the sample sizes were not powered for analyzing sex differences. Investigators were blinded to the genotype of mice during behavioral assays as well as throughout the data analysis. Computer code for data analysis is publicly available on Figshare (*Torres-Herraez and Martyniuk, 2022a*; *Torres-Herraez and Martyniuk, 2022b*).

## Acknowledgements

We thank Julia Greenwald for technical help. This application was funded by R01 MH124858, RO1 MH093672 to C.K., and D-Span award F99/K00 D-SPAN NS120642 to K.M.

## Additional information

### Funding

| Funder | Grant reference number | Author |
| --- | --- | --- |
| National Institute of Neurological Disorders and Stroke | F99/K00 D-SPAN NS120642 | Kelly M Martyniuk |
| National Institute of Mental Health | R01MH124858 | Christoph Kellendonk |
| National Institute of Mental Health | R01MH093672 | Christoph Kellendonk |
| European Molecular Biology Organization | Individual Fellowship | Arturo Torres-Herraez |

The funders had no role in study design, data collection and interpretation, or the decision to submit the work for publication.

## Author contributions

Kelly M Martyniuk, Conceptualization, Formal analysis, Investigation, Methodology, Writing – original draft, Data curation, Funding acquisition, Validation, Writing – review and editing, Supervision; Arturo Torres-Herraez, Software, Formal analysis, Writing – review and editing; Daniel C Lowes, Conceptualization, Formal analysis, Investigation, Writing – review and editing; Marcelo Rubinstein, Resources, Writing – review and editing; Marie A Labouesse, Investigation, Methodology, Writing – review and editing; Christoph Kellendonk, Conceptualization, Resources, Supervision, Funding acquisition, Methodology, Project administration, Writing – review and editing

## Author ORCIDs

Kelly M Martyniuk (iD) http://orcid.org/0000-0003-4190-4121
Marie A Labouesse (iD) http://orcid.org/0000-0002-6850-5852
Christoph Kellendonk (iD) http://orcid.org/0000-0003-3302-2188

## Ethics

all experimental procedures were conducted following NIH guidelines and were approved by Institutional Animal Care and Use Committees by Columbia University and the New York State Psychiatric Institute (NYSPI protocol #1621).

## Decision letter and Author response

Decision letter https://doi.org/10.7554/eLife.76111.sa1
Author response https://doi.org/10.7554/eLife.76111.sa2

---

# Additional files

## Supplementary files

• Transparent reporting form

## Data availability

All data generated or analyzed during this study are included in the manuscript.

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
