## [Editor Report]

The study addressed interactions between two key striatal transmitters dopamine and acetylcholine during an appetitive behavioral task. Helping to reconcile conflicting evidence in the literature, the data show that changes in both transmitters are correlated and that decreases in acetylcholine with reward and reward cues are only partially a consequence of elevated dopamine release acting at D2 dopamine receptors on striatal cholinergic interneurons. This manuscript will be of interest to those interested in the neural correlates of appetitive behavior and dopamine and striatal function.

---

## [Decision Letter]

**Decision letter after peer review:**

Thank you for submitting your article "Dopamine D2Rs Coordinate Cue-Evoked Changes in Striatal Acetylcholine Levels" for consideration by *eLife*. Your article has been reviewed by 3 peer reviewers, and the evaluation has been overseen by a Reviewing Editor and Kate Wassum as the Senior Editor. The following individuals involved in review of your submission have agreed to reveal their identity: Margaret Rice (Reviewer #1).

The reviewers have discussed their reviews with one another, and the Reviewing Editor has drafted this to help you prepare a revised submission. In your response to review please address each of the following essential points as well as the points raised by the reviewers in both their public and 'for authors' comments.

Essential revisions:

1) Please use the word "decrease" instead of "dip" throughout the manuscript. A "dip" reflects a decrease in ACh release and consequently detected ACh concentration. It is not the phenomenon that causes a change in behavior per se, as stated (e.g., li. 151).

2) Please provide the relative response times of sensors used for ACh and DA (and how this was assessed), as identical response (kinetic) parameters are essential for the temporal conclusions made, especially since the measurements were obtained in different locations.

3) It is unclear why there is a decrease in ACh in panel 1D but not 1C? Both responses are said to have been recorded at the time of level extension.

4) The use of scopolamine blocks detection of whatever the sensor is detecting; it does not alter ambient ACh levels, please modify the text accordingly (li. 320-325).

5) Local increases in ACh release regulate local DA release – rather than the apparent assumption of the authors that the increase in DA signal comes solely from increased activity of distal DA neurons. In support of this, some of the records (e.g., Figure 1D) show an increase in ACh levels before the larger decrease seen, although this is not seen in all instances (Figure 1C). Also, there was no change in dopamine release following CIN-D2 KO (Figure S3), despite a change in ACh release. Please explain why changes in ACh release did not affect dopamine release? This converse interaction (ACh promoting DA release) is not discussed, but addressing it would contribute to the on-going debate about local release processes vs. DA neurons firing rate in governing striatal DA release during behavior and strengthen the report.

6) It appears from figure 10 that ChATDrd2KO mice exhibit no change in press latency compared to controls. If no clear behavioral change occurs following CIN-D2 KO, the relevance of changes in ACh release are limited and may be more of an epiphenomenon. Please clarify.

7) Much of the analysis focuses on trials with press latencies > 2s. However, these results may be confounded by a change in trial number due to the pharmacological action of D2 receptor manipulation. This needs to be addressed with appropriate controls, which the authors should have.

8) The authors observe differences in Pearson's r between groups (Figures8 and 10 for example), but this difference is never statistically compared. Notably, the statistical analysis should use Fisher's r to z transformation because correlation coefficients are not an appropriate distribution for ANOVAs.

9)Please include a key resource table if you have not already done so.

*Reviewer #1 (Recommendations for the authors):*

This report addresses a long-standing question, which is the extent to which the cholinergic "pause" is due to D2 dopamine (DA) receptor activation on cholinergic interneurons (ChIs). Most of the studies, including pharmacological blockade of D2 receptors, and, more importantly, selective deletion of D2 receptors in ChIs. The paper convincingly shows that D2 receptors regulate the duration of the reward- or cue-associated decrease in ACh levels monitored with a genetically encoded sensor, and that this regulation contributes to the latency to press a level for reward. This is a strong contribution to the literature. But the paper then goes on to examine correlations between DA and ACh levels during various aspects of the behavior. The clarity of this section could be improved. More importantly, there is no consideration of whether ACh might promote DA release at any phase of the behaviors examined. This seems an oversight, that may reflect the authors' perspective that striatal DA release is governed by midbrain DA neuron firing only. Although examining DA and ACh in separate hemispheres may preclude addressing this, evidence may be found in some of the many correlations reported.

1) Abstract. The word "decrease" would be better than "dip" here (and in my opinion, throughout the paper); if used, dip "length" would be better as dip "duration". Elsewhere, dip is more nearly defined and is OK to use, although it comes across as jargon. A "dip" reflects a decrease in ACh release and consequently detected ACh concentration. It is not the thing that causes a change in behavior per se, as stated (e.g., li. 151).

2) The relative response times of sensors used for ACh and DA need to be well-described (including how this was assessed), as identical response parameters are essential for the temporal conclusions made, especially since the measurements were in different locations.

3) Figure 1. Why is there a peak for ACh in 1D but not 1C? Both responses are said to have been recorded at the time of level extension. What was the relative percent of time each pattern was observed in there studies?

4) Use of scopolamine blocks detection of whatever the sensor is detect; it does not abolish changes in ACh levels (li. 320-325).

5) Figure 1E. The figure implies that the duration of the observed decrease in ACh levels with level extension lengthens progressively with trial number. However, the legend says that the responses were from a number of trials in a number of mice, and were sorted based on "length of ACh decrease" for 300 trials. In this light, what the panel is supposed to show is not clear.

6) The figures in which a D2 antagonist is administered should have the drug name included in the x-axis of the related figures, not just the concentration.

7) 334-335. Certainly, ACh release is regulated by CIN activity, but it seems unlikely that this also controls degradation or diffusion of ACh after release. This needs to be reworded to be accurate.

8) 589-590. Demonstration of neuronal responses to reward has been reported, so saying that the present data confirm this is overstated. Better would be to say that the results are in keeping with previous studies.

9) A general omission in the paper is examination of the possibility that local increases in ACh release might regulate local DA release – rather than the apparent assumption of the authors that the increase in DA signal comes only from increased activity of distant DA neurons. In support of this, some of the records (e.g., Figure 1D) show an increase in ACh levels before the larger decrease seen, although this is not seen in all (Figure 1C). This converse interaction (ACh promoting DA release) is not discussed, but addressing it would contribute to the on-going debate about local release processes vs. DA neurons firing rate in governing striatal DA release and strengthen the report.

10) 688-680. The authors say that a "positive correlation between the size of the stimulus induced ACh dip and behavioral responding, which implicates a role for ACh in motivated behaviors." However, they show a role for DA acting at D2 receptors in the "dip", so that it seems equally likely that this observation reflects increased dopamine release and its effects on behavioral responding.

*Reviewer #2 (Recommendations for the authors):*

1. The behavioral relevance of CIN-D2 receptor control of ACh release is not clear.

a. How did eticlopride or CIN-D2 KO alter press latency, head entry latency, total number of trials completed, etc? It appears from figure 10 that ChATDrd2KO mice exhibit no change in press latency compared to controls. If no clear behavioral change occurs following CIN-D2 KO, the relevance of changes in ACh release are extremely limited and may be nothing more than an epiphenomenon.

b. Importantly, did these manipulations alter the proportion of trials with press latencies > 2s? Because much of the analysis focuses on trials with press latencies > 2s, results may be confounded by a change in trial number due to D2 receptor manipulation. Please clarify.

2. Extensive work demonstrates that striatal CINs locally modulate terminal dopamine release. However, there does not appear to be any change in dopamine release following CIN-D2 KO (Figure S3), despite a change in ACh release. Please explain why changes in ACh release did not affect dopamine release?

3. The authors interpret several comparisons between groups without directly comparing them. For example, in Figure 8, the authors compare the eticlopride effects in Drd2fl/fl and ChATDrd2KO mice and note that the effect is absent in ChATDrd2KO mice. A somewhat similar statistical error arises for the data presented in Figure 10. The authors observe differences in Pearson's r between groups, but this difference is never statistically compared. Notable, the statistical analysis should use Fisher's r to z transformation because correlation coefficients are not an appropriate distribution for ANOVAs.

*Reviewer #3 (Recommendations for the authors):*

1. Compare the press latency between control and ChAT-Drd2KO mice.

2. Please show if there is any correlation between ACh AUC and other behavior indexes such as press speed or the time between press and reward licking.

[Editors' note: further revisions were suggested prior to acceptance, as described below.]

Thank you for resubmitting your work entitled "Dopamine D2Rs Coordinate Cue-Evoked Changes in Striatal Acetylcholine Levels" for further consideration by *eLife*. Your revised article has been evaluated by Kate Wassum (Senior Editor) and a Reviewing Editor.

The manuscript has been improved but there are some remaining issues that need to be addressed, as outlined below:

While the reviewers think that the authors were very responsive to the initial round of review, some important questions remain regarding the statistical power of some of the experiments. Reviewers request that the authors refrain from stating that there are any age or sex differences as the groups lack the required sample size to obtain statistical veracity to those claims. Please also include the precise details of the power analysis used to justify that each experiment was adequately powered. This is especially important for cases where the N's are 4-5.

---

## [Author Response]

Essential revisions:1) Please use the word "decrease" instead of "dip" throughout the manuscript. A "dip" reflects a decrease in ACh release and consequently detected ACh concentration. It is not the phenomenon that causes a change in behavior per se, as stated (e.g., li. 151).

We changed “dip” to “decrease” where possible.

2) Please provide the relative response times of sensors used for ACh and DA (and how this was assessed), as identical response (kinetic) parameters are essential for the temporal conclusions made, especially since the measurements were obtained in different locations.

We now added the kinetics of each sensor in the methods section and a paragraph in the discussion addressing sensor kinetics within our findings (Line 494):

“Electrical stimulation in brain slices revealed on/off kinetics for dLight1.2 of 9.5 ms and 90 ms, respectively (Labouesse and Patriarchi, 2021). In contrast, the kinetics for the ACh3.0 sensor after electrical stimulation were slower with on/off kinetics of 90-105 ms and 0.9-3.7 s, respectively (Jing et al., 2020). Here, we measured a decrease in striatal ACh levels in vivo with an onset of 206 ms after optogenetic inhibition of CINs. While this is significantly shorter than the reported on/off kinetics for the ACh3.0 sensor, it is possible that the lag of the cue induced decrease in ACh in relationship to the peak in DA is due to differences in the on/off kinetics of the two sensors and not a natural reflection of when both events occur.”

3) It is unclear why there is a decrease in ACh in panel 1D but not 1C? Both responses are said to have been recorded at the time of level extension.

We believe that the reviewer is asking why there is an initial peak in ACh levels in 1D but not 1C. This is a great question, and we don’t really know why this is the case but believe that this could be due to differences in the fiber location. We have observed the peak in some, but not other mice. We found that C57BL/6J had a smaller initial ACh peak compared to Drd2^fl/fl^ control and ChATDrd2KO mice, We added a figure to the supplementary figures (Figure 1-S1) showing the optical fiber locations and example traces for individual mice. We found that the optical fibers in C57BL/6J mice were more ventral and medial (Figure 1-S1A), while in the Drd2^fl/fl^ (Figure 1-S1B) and ChATDrd2KO (Figure 1-S1C) they were slightly more dorsal and lateral. Therefore, the ACh initial peak may be due to location of where we were recording ACh levels.

We added the following paragraph to the result section (line 184):

“Variability between animals:

While analyzing the ACh signals we found that some mice showed an initial peak in ACh levels while other did not. This has been described at the level of neuronal when recording from individual neurons (Apicella et al., 1997; Kimura et al., 1984) but here it is observed at the level of ACh levels released by a population of neurons. While the origin of the between animal variability is unclear, we believe that it is related to the location of recording. Generally, more lateral/dorsal recording location showed an initial peak in ACh levels while more medial/ventral location did not show the initial peak (Figure 1-S1). The origin for this variability should be addressed in a more systemic way in the future.”

4) The use of scopolamine blocks detection of whatever the sensor is detecting; it does not alter ambient ACh levels, please modify the text accordingly (li. 320-325).

We modified the text to reflect that scopolamine is blocking what the sensor is detecting (Line 168).

“We found that the competitive antagonist scopolamine abolished the early increase and the subsequent decrease in the fluorescent signal, indicating that GACh3.0 indeed quantifies ACh binding and thus surrounding ACh levels (Figure 1D).”

5) Local increases in ACh release regulate local DA release – rather than the apparent assumption of the authors that the increase in DA signal comes solely from increased activity of distal DA neurons. In support of this, some of the records (e.g., Figure 1D) show an increase in ACh levels before the larger decrease seen, although this is not seen in all instances (Figure 1C). Also, there was no change in dopamine release following CIN-D2 KO (Figure S3), despite a change in ACh release. Please explain why changes in ACh release did not affect dopamine release? This converse interaction (ACh promoting DA release) is not discussed, but addressing it would contribute to the on-going debate about local release processes vs. DA neurons firing rate in governing striatal DA release during behavior and strengthen the report.

This is an important point, and we added a paragraph in the discussion addressing this point (Line 469):

“Slice and in vivo stimulation studies have shown that stimulation of ACh release induces DA release via activation of nicotinic receptors on DA terminals (Cachope and Cheer, 2014; Cachope et al., 2012; Sulzer et al., 2016; Threlfell et al., 2012). Thus, it is surprising that despite a change in cue induced ACh signal the cue-induced DA signal is largely unchanged. Especially, we hypothesized that the rebound in ACh levels observed in the eticlopride conditions may enhance or prolong the DA peak. Although the early induction of the DA peak is indeed enhanced in the eticlopride condition this is also observed in CIN-selective D2R KO mice. We believe that the enhanced initial peak in the eticlopride condition is due to antagonism of D2 auto receptors on DA terminals. It is unclear why there is no additional effect on DA release as a consequence of the ACh rebound but it could be that under the behavioral conditions of the CRF task nicotinic receptors on DA neurons are desensitized.”

6) It appears from figure 10 that ChATDrd2KO mice exhibit no change in press latency compared to controls. If no clear behavioral change occurs following CIN-D2 KO, the relevance of changes in ACh release are limited and may be more of an epiphenomenon. Please clarify.

To address the importance of cue-induced changes in ACh in the regulation of press latencies we inhibited CIN-neurons at the time of lever extension. We found that a long inhibition of CINs resulted in a decreased press latency when compared to a short inhibition CINs. We added this data to the Results section (Figure 11). We added the following paragraph to the Results section (Line 373):

“ACh decrease duration regulates motivated behavior

To determine whether ACh decrease duration controls motivated behavior, we expressed GACh3.0 and Cre-dependent eNpHR3.0 in the DMS of ChAT-IRES-Cre mice and trained these mice on the CRF task. After mice achieved criterion performance, we recorded GACh3.0 signals during the CRF task as above. On a subset of trials in each session, we inhibited CINs at lever extension for a short (0.5 s + 1 s ramp down) or long (0.5 s + 3 s ramp down) duration to produce short and long duration decreases in ACh levels (Figure 11A). Consistent with the correlation we observed between ACh decrease size and lever press latency, we found lever press latency was longer during short inhibition trials compared to long inhibition trials (Figure 11B). These data are consistent with the change in cue-induced ACh levels regulating motivated behavior.”

We added the following paragraph to the discussion (line 517 and 540)

“To address whether the duration of cue induced decrease in ACh levels alter press latencies, we used optogenetic tools to artificially induce shorter and longer decreases in ACh at cue presentation. We found that short inhibition of CINs increased press latency compared to long inhibition of CINs. However, the longer inhibition also led to a larger rebound and both changes may be responsible for the change in press latency. Nevertheless, this suggests that the cue-evoked change in ACh levels regulate the motivation to initiate the lever press.”

“Surprisingly, we did not measure an increase in press latencies in D2R KO mice, despite the shortened dip duration. This discrepancy may be due to compensation in the regulation of press latencies via other mechanisms that have developed during development or as the consequence of the chronic change in D2R levels. Such compensatory mechanisms may not have the time to develop in response to the acute action of the antagonist. Alternatively, the correlation between cue induced dip-duration and press latency could be an epiphenomenon. For example, D2R antagonism may change both, tonic ACh levels (Ikarashi et al., 1997) and cue induced changes in ACh, but it is only the first one that regulates press latencies. The optogenetic inhibition experiment argues otherwise as the short inhibition of CINs increased press latency in comparison to long inhibition of CINs, suggesting that differences in phasic ACh levels are sufficient to regulate motivated behavior. Further investigation is needed to determine whether compensatory mechanisms overcome changes in cue induced changes in ACh levels in D2R KO mice to normalize motivated behavior.”

7) Much of the analysis focuses on trials with press latencies > 2s. However, these results may be confounded by a change in trial number due to the pharmacological action of D2 receptor manipulation. This needs to be addressed with appropriate controls, which the authors should have.

We plotted the data for all trials and found similar results as trials with press latencies > 2 s and obtained the same results. We did not present the data in the original submission to not overload the reader with too many figures. We are happy to show the data to the reviewers in Author response images 1-3 and can include that data as a supplementary figure if requested.

**Author response image 1. sa2fig1:** Selective D2R ablation from CINs alters the cue evoked ACh pause when using all trials. (A) Changes in ACh fluorescence (ΔF/F (%)) aligned to lever extension for all trials for control (black) and D2R KO (blue), N=4/ genotype. (B) There is no difference in the dip amplitude between Drd2fl/fl control and ChATDrd2KO mice (p = 0.2289). (C) The dip duration is significantly shorter in D2R KO animals compared to controls (p = 0.0213).

**Author response image 2. sa2fig2:** D2R antagonism alters the cue evoked ACh dip in Drd2fl/fl control mice when using all trials (A) Changes in ACh fluorescence (ΔF/F (%)) aligned to lever extension for all trials with increasing doses of eticlopride. (B) Negative AUC is decreased by eticlopride in a dose-dependent manner (F(1.236, 3.708) = 13.65, p = 0.0224). (C) Rebound AUC is increased by eticlopride in a dose-dependent manner (F(1.546, 4.639) = 8.457, p = 0.0312). (D) Total AUC is increased dose-dependently by eticlopride (F(1.122, 3.365) = 8.852, p = 0.0050). (E) Dip duration is decreased by eticlopride in a dose-dependent manner (F(1.080, 3.241) = 10.74, p = 0.0411). (F) Dip amplitude is not affected by eticlopride (F(1.135, 3.405) = 7.999, p = 0.0562).

**Author response image 3. sa2fig3:** D2R antagonism does not the cue evoked ACh dip in ChATDrd2KO mice when using all trials (A) Changes in ACh fluorescence (ΔF/F (%)) aligned to lever extension for all trials with increasing doses of eticlopride. (B) Negative AUC is not affected by eticlopride (F(1.404, 4.213) = 1.248, p = 0.3507). (C) Rebound AUC is not affected by eticlopride (F(2.414, 7.243) = 2.689, p = 0.1296). (D) Total AUC is not affected by eticlopride (F(1.844, 5.532) = 1.079, p = 0.3958). (E) Dip duration is not affected by eticlopride (F(1.701, 5.103) = 0.7879, p = 0.4835). (F) Dip amplitude is not affected by eticlopride (F(1.601, 4.803) = 3.232, p = 0.1316).

8) The authors observe differences in Pearson's r between groups (Figures8 and 10 for example), but this difference is never statistically compared. Notably, the statistical analysis should use Fisher's r to z transformation because correlation coefficients are not an appropriate distribution for ANOVAs.

We performed the Fisher’s r to z transformation and obtained similar results. Therefore, we changed the analysis to show the Fisher’s r to z transformation in Figures 5, 6, 9, 9-S1, 9-S2, 9-S3 and 9-S4.

9)Please include a key resource table if you have not already done so.

A key resource table was added to the beginning of the methods section.

Reviewer #1 (Recommendations for the authors):This report addresses a long-standing question, which is the extent to which the cholinergic "pause" is due to D2 dopamine (DA) receptor activation on cholinergic interneurons (ChIs). Most of the studies, including pharmacological blockade of D2 receptors, and, more importantly, selective deletion of D2 receptors in ChIs. The paper convincingly shows that D2 receptors regulate the duration of the reward- or cue-associated decrease in ACh levels monitored with a genetically encoded sensor, and that this regulation contributes to the latency to press a level for reward. This is a strong contribution to the literature. But the paper then goes on to examine correlations between DA and ACh levels during various aspects of the behavior. The clarity of this section could be improved. More importantly, there is no consideration of whether ACh might promote DA release at any phase of the behaviors examined. This seems an oversight, that may reflect the authors' perspective that striatal DA release is governed by midbrain DA neuron firing only. Although examining DA and ACh in separate hemispheres may preclude addressing this, evidence may be found in some of the many correlations reported.1) Abstract. The word "decrease" would be better than "dip" here (and in my opinion, throughout the paper); if used, dip "length" would be better as dip "duration". Elsewhere, dip is more nearly defined and is OK to use, although it comes across as jargon. A "dip" reflects a decrease in ACh release and consequently detected ACh concentration. It is not the thing that causes a change in behavior per se, as stated (e.g., li. 151).

We changed “dip” to “decrease” where necessary.

2) The relative response times of sensors used for ACh and DA need to be well-described (including how this was assessed), as identical response parameters are essential for the temporal conclusions made, especially since the measurements were in different locations.

See response to public comment number 2.

3) Figure 1. Why is there a peak for ACh in 1D but not 1C? Both responses are said to have been recorded at the time of level extension. What was the relative percent of time each pattern was observed in there studies?

See response to public comment number 3.

4) Use of scopolamine blocks detection of whatever the sensor is detect; it does not abolish changes in ACh levels (li. 320-325).

See response to public comment number 4.

5) Figure 1E. The figure implies that the duration of the observed decrease in ACh levels with level extension lengthens progressively with trial number. However, the legend says that the responses were from a number of trials in a number of mice, and were sorted based on "length of ACh decrease" for 300 trials. In this light, what the panel is supposed to show is not clear.

Figure 1E is supposed to show that there are varying lengths of the measured decreases in ACh levels. We added the title, “Trials sorted by duration of the decrease in ACh levels” to Figure 1E.

6) The figures in which a D2 antagonist is administered should have the drug name included in the x-axis of the related figures, not just the concentration.

We now added the drug name to the x-axis labels where necessary.

7) 334-335. Certainly, ACh release is regulated by CIN activity, but it seems unlikely that this also controls degradation or diffusion of ACh after release. This needs to be reworded to be accurate.

We now clarified this statement in the text (Line 181):

“It also indicates that ACh levels are tightly controlled by CIN neuron activity.”

8) 589-590. Demonstration of neuronal responses to reward has been reported, so saying that the present data confirm this is overstated. Better would be to say that the results are in keeping with previous studies.

A valid point. We reworded this statement in the text (Line 429):

“This is consistent with previous studies showing that DA neurons and CINs respond to salient and conditioned stimuli.”

9) A general omission in the paper is examination of the possibility that local increases in ACh release might regulate local DA release – rather than the apparent assumption of the authors that the increase in DA signal comes only from increased activity of distant DA neurons. In support of this, some of the records (e.g., Figure 1D) show an increase in ACh levels before the larger decrease seen, although this is not seen in all (Figure 1C). This converse interaction (ACh promoting DA release) is not discussed, but addressing it would contribute to the on-going debate about local release processes vs. DA neurons firing rate in governing striatal DA release and strengthen the report.

See response to public comment number 5.

10) 688-680. The authors say that a "positive correlation between the size of the stimulus induced ACh dip and behavioral responding, which implicates a role for ACh in motivated behaviors." However, they show a role for DA acting at D2 receptors in the "dip", so that it seems equally likely that this observation reflects increased dopamine release and its effects on behavioral responding.

We quantified DA signal (max peak and AUC) and behavioral responding. We found no correlation between the DA max peak and press latency or DA AUC (B) and press latency. We added the following sentences to the Results section (line 366):

“In addition, we analyzed the correlation between DA and behavioral responding and found no correlation between the DA AUC or max peak and lever press latency (Figure 10-S1). This result suggests that DA does not affect behavioral responding in this CRF task.”

Reviewer #2 (Recommendations for the authors):1. The behavioral relevance of CIN-D2 receptor control of ACh release is not clear.a. How did eticlopride or CIN-D2 KO alter press latency, head entry latency, total number of trials completed, etc? It appears from figure 10 that ChATDrd2KO mice exhibit no change in press latency compared to controls. If no clear behavioral change occurs following CIN-D2 KO, the relevance of changes in ACh release are extremely limited and may be nothing more than an epiphenomenon.

Press latency: See response to public comment number 6.

Head entry latency: We found that eticlopride increased the latency to retrieve the reward in C57BL/6J (wildtype) and Drd2^fl/fl^ control but not ChATDrd2KO mice. Like for press latency there was no difference in the latency to retrieve the reward between Drd2^fl/fl^ and ChATDrd2KO mice at baseline (saline) (see Author response image 4).

**Author response image 4. sa2fig4:** Head entry latency. Latency to head entry for reward after lever press with increasing doses of eticlopride (A) Drd2fl/fl control mice (F(1.223, 3.670) = 7.677, p = 0.0535) or (B) ChatDrd2KO mice (F(1.106, 3.317) = 1.483, p = 0.3100) (C) No difference in head entry latency between Drd2fl/fl and ChATDrd2 KO mice at baseline (p = 0.5805).

Total trials completed: We also determined that eticlopride significantly reduced the total # of trials completed in both Drd2^fl/fl^ control mice and ChATDrd2KO mice. The effect in ChATDrd2KO mice may be due to blocking D2Rs on indirect pathway projection neurons. There was no difference in the total # of trials completed between the two groups at baseline (saline) (see Author response image 5). We added the trial completion analysis to the manuscript (Figure 10C).

**Author response image 5. sa2fig5:** Trials Completed. The total # of trials completed was significantly reduced by eticlopride in (A) Drd2fl/fl control mice (F(1.607, 4.820) = 35.14, p = 0.0015) or (B) ChatDrd2KO mice (F(1.649, 4.948) = 7.371 p = 0.0353) (C) No difference in the # of completed trials between Drd2fl/fl and ChATDrd2 KO mice at baseline.

b. Importantly, did these manipulations alter the proportion of trials with press latencies > 2s? Because much of the analysis focuses on trials with press latencies > 2s, results may be confounded by a change in trial number due to D2 receptor manipulation. Please clarify.

See answer to public comment 7.

2. Extensive work demonstrates that striatal CINs locally modulate terminal dopamine release. However, there does not appear to be any change in dopamine release following CIN-D2 KO (Figure S3), despite a change in ACh release. Please explain why changes in ACh release did not affect dopamine release?

See answer to public comment 5.

3. The authors interpret several comparisons between groups without directly comparing them. For example, in Figure 8, the authors compare the eticlopride effects in Drd2fl/fl and ChATDrd2KO mice and note that the effect is absent in ChATDrd2KO mice. A somewhat similar statistical error arises for the data presented in Figure 10. The authors observe differences in Pearson's r between groups, but this difference is never statistically compared. Notable, the statistical analysis should use Fisher's r to z transformation because correlation coefficients are not an appropriate distribution for ANOVAs.

A valid comment. We compared the effects of eticlopride between Drd2^fl/fl^ and ChATDrd2KO mice and determined that there is a genotype x dose interaction on all measures quantified in Figure 8 (see Author response image 6).

**Author response image 6. sa2fig6:** Genotype x dose effect of eticlopride between Drd2fl/fl and ChatDrd2KO mice. (A) A trending but non-significant effect on the Negative AUC (Genotype x dose: F(3, 18) = 3.113, p = 0.0522). (B) A significant effect on the Rebound AUC (Genotype x dose: F(3,18) = 4.600, p=0.0147). (C) A significant effect on the Total AUC (Genotype x dose: F(3,18) = 8.106, p=0.0013). (D) A significant effect on the Dip Duration (Genotype x dose: F(3,18) = 10.41, p=0.0003). (E) No effect on the Dip Amplitude (Genotype x dose: F(3,18) = 1.611, p=0.2219).

We added the following paragraph to the result section (line 306):

“When Drd2^fl/fl^ and ChATDrd2KO mice were analyzed together we measured a gene x eticlopride interaction for negative AUC (Genotype x dose: F_(3, 18)_ = 3.113, p = 0.0522), rebound AUC (Genotype x dose: F_(3,18)_ = 4.600, p=0.0147), total AUC (Genotype x dose: F_(3,18)_ = 8.106, p=0.0013), dip Duration (Genotype x dose: F_(3,18)_ = 10.41, p=0.0003) but not dip Amplitude (Genotype x dose: F_(3,18)_ = 1.611, p=0.2219).”

We performed the Fisher’s r to z transformation and obtained similar results. Therefore, we changed the analysis to show the Fisher’s r to z transformation in Figures 5, 6, 9, 9-S1, 9-S2, 9-S3 and 9-S4.

Reviewer #3 (Recommendations for the authors):1. Compare the press latency between control and ChAT-Drd2KO mice.

There is no difference in press latency between controls and KOs at baseline (Figure 10B).

2. Please show if there is any correlation between ACh AUC and other behavior indexes such as press speed or the time between press and reward licking.

See response to the comment above.

[Editors' note: further revisions were suggested prior to acceptance, as described below.]

The manuscript has been improved but there are some remaining issues that need to be addressed, as outlined below:While the reviewers think that the authors were very responsive to the initial round of review, some important questions remain regarding the statistical power of some of the experiments. Reviewers request that the authors refrain from stating that there are any age or sex differences as the groups lack the required sample size to obtain statistical veracity to those claims.

We removed the comment addressing ages and sex from the method section and replaced it with the following sentence in line 729:

“Both male and female mice were analyzed, however, the sample sizes were not powered for analyzing sex differences.”

Please also include the precise details of the power analysis used to justify that each experiment was adequately powered. This is especially important for cases where the N's are 4-5.

We included the following power analysis in line 710 of the methods section:

To determine the sample size for detecting whether D2R antagonism alters the cue-evoked change in ACh levels we performed a power analysis on the first cohort of wild-type mice using total AUC from Fig. 4 as the outcome measure. The analysis was performed in Gpower using the repeated measures, within factors ANOVA power analysis. In a first step we calculated a partial eta square using F*df_effect / (F*df_effect + df_error which then was used to calculate the effects size f. A non-sphericity correction epsilon was calculated in Prism GraphPad to quantify how variances differ between different repeated measures. A correlation matrix between repeated measures was used to determine the average correlation between repeated measures. Based on this we calculated for AUC (eta square: 0.681, effect size f: 1.46, alpha: 0.05, Power: 0.8, repeated measures: 6, corr among repeated measures: -0.068), non-sphericity correction: 0.3193) a sample size of n=5. A similar analysis for press latency (Input: eta square: 0.620, effect size f: 1.28, alpha: 0.05, Power: 0.8, repeated measures: 6, corr among repeated measures: 0.1294, non-sphericity correction: 0.278) revealed a sample size of n=5. Expecting similar effect sizes in follow up experiments we aimed for a sample size of 5 mice per group when analyzing the CIN-selective D2R KO mice and their littermate controls.